# BLACK-BOX DETECTION OF LANGUAGE MODEL WATERMARKS

**Thibaud Gloaguen, Nikola Jovanović, Robin Staab, Martin Vechev**
ETH Zurich
tgloaguen@ethz.ch, {nikola.jovanovic, robin.staab, martin.vechev}@inf.ethz.ch

## ABSTRACT

Watermarking has emerged as a promising way to detect LLM-generated text, by augmenting LLM generations with later detectable signals. Recent work has proposed multiple families of watermarking schemes, several of which focus on preserving the LLM distribution. This distribution-preservation property is motivated by the fact that it is a tractable proxy for retaining LLM capabilities, as well as the inherently implied undetectability of the watermark by downstream users. Yet, despite much discourse around undetectability, no prior work has investigated the practical detectability of any of the current watermarking schemes in a realistic black-box setting. In this work we tackle this for the first time, developing rigorous statistical tests to detect the presence, and estimate parameters, of all three popular watermarking scheme families, using only a limited number of black-box queries. We experimentally confirm the effectiveness of our methods on a range of schemes and a diverse set of open-source models. Further, we validate the feasibility of our tests on real-world APIs. Our findings indicate that current watermarking schemes are more detectable than previously believed.

## 1 INTRODUCTION

With the rapid increase in large language model (LLM) capabilities and their widespread adoption, researchers and regulators alike are raising concerns about their potential misuse for generating harmful content (Bommasani et al., 2021; EU Council, 2024). Tackling this issue, LLM watermarking, a process of embedding a signal invisible to humans into generated texts, gained significant traction.

**Language model watermarking** More formally, in LLM watermarking (Hu et al., 2024; Kirchenbauer et al., 2023; 2024; Kuditipudi et al., 2024; Sadasivan et al., 2023; Wang et al., 2024; Wu et al., 2024) we consider the setting of a *model provider* that offers black-box access to their proprietary model $LM$ while ensuring that each generation $y$ in response to a prompt $q$ can be reliably attributed to the model. To enable this, the provider modifies the generations using a secret watermark key $\xi$, that a corresponding watermark detector can later use to detect whether a text was generated by $LM$.

The prominent family of *Red-Green* watermarks (Kirchenbauer et al., 2023; 2024) achieves this by, at each step of generation, selecting a context-dependent pseudorandom set of logits to be boosted, modifying the model's next-token distribution. In contrast, the recently proposed families of *Fixed-Sampling* (Kuditipudi et al., 2024) and *Cache-Augmented* (Hu et al., 2024; Wu et al., 2024) schemes have focused on developing watermarks that aim to preserve the output distribution of the LLM. Both families achieve these guarantees in ideal theoretical settings, but no previous work has studied whether implementation constraints may violate those guarantees.

**Watermark detectability** The other key motivation of these distribution-preserving watermarks has been their inherently implied *undetectability*, which ideally makes it *"impossible for users to discern [whether a watermark has been applied]"* (Hu et al., 2024), resulting in a *"stealthy watermark, [i.e., one whose presence is hard to distinguish from an unwatermarked LM via sampling]"* (Wu et al., 2024). Such focus on undetectability in prior work raises the following research question:

*"How detectable are existing watermarks in practical black-box settings?"*

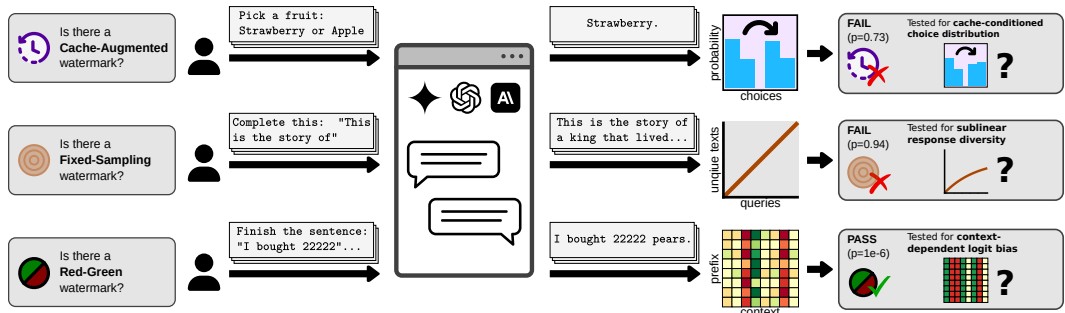

Figure 1: Overview of our key contribution. Given black-box textual access to a language model, a client can query the model and run statistical tests to rigorously test for presence of a watermark. In this example, both the test for *Cache-Augmented* watermarks (§4) and the test for *Fixed-Sampling* watermarks (§3) fail, while the test for *Red-Green* watermarks (§2) successfully detects the watermark.

Surprisingly, while this question concerns an increasingly popular property of LLM watermarks, and was raised as practically relevant in prior works such as Piet et al. (2023) and Christ et al. (2024), it has remained unanswered for any of the most popular families of LLM watermarking schemes.

**This work: practical black-box watermark detection**    In this work, for the first time, we comprehensively study the question of watermark detection in practical real-world settings. Faithful to how LLMs are exposed in current public deployments, we assume a minimal setting in which the adversary cannot control any input parameters (e.g., temperature, sampling strategy) apart from the prompt $q$, and does not receive any additional information (e.g., log-probabilities) apart from the instruction-tuned model's textual response $y$. In this setting, the goal of the adversary is to identify if the model is watermarked and determine which scheme was used. As shown in recent work, this information can also enable stronger attacks on the watermark such as removal or spoofing (Jovanović et al., 2024; Pang et al., 2024; Sadasivan et al., 2023; Zhang et al., 2024). We explore the practical implications of watermark detection in App. A.

To achieve this goal, we propose rigorous and principled statistical tests for black-box detection of seven scheme variants of the three most prominent watermark families. Our tests, illustrated in Fig. 1, are based on fundamental properties of the three respective scheme families: Red-Green (§2), Fixed-Sampling (§3), and Cache-Augmented (§4) watermarks. In our extensive experimental evaluation, we find that in practice, both distribution-modifying as well as distribution-preserving schemes are *easily detectable* by our tests, even in the most restricted black-box setting. Highlighting the practicality of our tests, we show how they can be directly applied to several real-world black box LLM deployments (GPT4, CLAUDE 3, GEMINI 1.0 PRO) at very little cost.

**Main contributions**    We make the following key contributions:

- We present the first principled and statistically rigorous tests for practical black-box detection of popular LLM watermarks across the three prominent scheme families: Red-Green (§2), Fixed-Sampling (§3), and Cache-Augmented (§4).

- We confirm the effectiveness and applicability of all our tests in an extensive experimental evaluation across seven schemes and five open-source models (§5.1 and §5.2) and execute them on three deployed models (§5.3).

- We develop tests to estimate important parameters of identified schemes (App. B), and further verify the robustness of our tests to additional scheme variants and adversarial modifications (App. F). Our code is publicly available at https://github.com/eth-sri/watermark-detection.

We believe this work is a promising step in evaluating the practical detectability of current LLM watermarking schemes in the realistic black-box setting.

## 2 DETECTING RED-GREEN WATERMARKS

In this section, we introduce our statistical test for detecting watermarks from the *Red-Green* family, illustrated in Fig. 1 (bottom). To this end, we exploit their core property: applying the watermark introduces a *context-dependent logit bias*, i.e., the model's output distribution is biased in a way that can greatly vary depending on the last few tokens that were generated (*context*), yet is unaffected by the tokens preceding the context (*prefix*). We provide a more formal description of the test in App. C.

We first recall the background related to these schemes. We then introduce our modeling assumptions, describe the querying strategy we use to obtain the data for the test, and finally describe the test itself. In §5, we experimentally confirm the effectiveness of the test in realistic settings and study its query cost, and in App. B.1 explore methods to estimate scheme parameters once a watermark is confirmed.

**Background** Assume a watermark key $\xi \in \mathbb{N}$, a pseudorandom function $f$, and a hashing function $Hash: \mathbb{N} \to \mathbb{N}$. At each generation step $t$, a Red-Green watermark modifies the logits $l_{1:|V|}$ of tokens from the vocabulary $V$ to promote a subset of tokens (*green tokens*) before applying standard sampling. We consider two popular variants, LEFTHASH (Kirchenbauer et al., 2023) and SELFHASH (Kirchenbauer et al., 2024), both parametrized by $\delta, \gamma \in \mathbb{R}^+$, and using $h = 1$ and $h = 3$ previous tokens as context, respectively. LEFTHASH seeds $f$ with $Hash(y_{t-1}) \cdot \xi$, and uses it to split $V$ into $\gamma|V|$ green tokens and remaining *red* tokens. For each green $i$, it then increases $l_i$ by $\delta$. SELFHASH differs by seeding $f$ with $\min(Hash(y_{t-3}), \ldots, Hash(y_t)) \cdot Hash(y_t) \cdot \xi$, effectively using a context size of 4 by including the token $y_t$ yet to be generated. For both schemes, the watermark detector is based on observing a significant number of green tokens in the input text. Other schemes from this family include varying the aggregation function or the context size (see §6).

**Modeling Red-Green watermarks** Assume a setting where the model chooses a token from some restricted set $\Sigma \subset V$, following some *context* $t_2$ (longer than the watermark context $h$), which is preceded by a *prefix* $t_1$. We discuss how to construct such a setting shortly. To model the watermark, we assume the following distribution for $p_{t_1,t_2}(x)$, the probability the model assigns to some $x \in \Sigma$:

$$p_{t_1,t_2}(x) = \frac{\exp\left((x^0 + \delta_{t_2}(x) + \varepsilon_{t_1,t_2}(x))/T\right)}{\sum_{w \in \Sigma} \exp\left((w^0 + \delta_{t_2}(w) + \varepsilon_{t_1,t_2}(w))/T\right)}, \tag{1}$$

where $T > 0$ is the sampling temperature. Here, we assume the existence of the *true logit* $x^0$, modeling the model bias towards $x$. The true logit is shifted by $\delta_{t_2}(x)$, a $\delta$-Bernoulli random variable, where $\delta \in \mathbb{R}^+$ is the watermark parameter introduced above. Finally, $\varepsilon_{t_1,t_2}(x)$ for different $t_1, t_2$ are iid symmetric error terms with mean 0 and variance $\sigma^2$. Applying the *logit* function $p \to \log(p/(1 - p))$ to Eq. (1), we obtain:

$$l_{t_1,t_2}(x) = \frac{x^0}{T} + \frac{\delta_{t_2}(x)}{T} + \frac{\varepsilon_{t_1,t_2}(x)}{T} - \log\left(\sum_{w \in \Sigma \setminus \{x\}} \exp\left(\frac{w^0}{T} + \frac{\delta_{t_2}(j)}{T} + \frac{\varepsilon_{t_1,t_2}}{T}\right)\right). \tag{2}$$

Approximating the log-sum-exp with a maximum, and WLOG setting $w^0 = 0$ for $w$ which is the maximum in the log-sum-exp (as logits are shift-invariant), the above simplifies to

$$l_{t_1,t_2}(x) = x^0/T + \delta'_{t_2}(x) + \varepsilon'_{t_1,t_2}(x), \tag{3}$$

where $\varepsilon'_{t_1,t_2}(x)$ absorbs the previous error terms, and $\delta'_{t_2}(x)$ is a random variable equal to 0 for un-watermarked model and defined in Eq. (4) for watermarked models, with $\delta$ the watermark parameter.

$$\delta'_{t_2}(x) = \begin{cases} \delta/T & \text{if } x \in G \text{ and } \forall y \in \Sigma, y \neq x \implies y \in R \\ -\delta/T & \text{if } x \in R \text{ and } \exists y \in \Sigma, y \in G \\ 0 & \text{otherwise.} \end{cases} \tag{4}$$

Our test is based on detecting cases of $\delta'_{t_2}(x) \neq 0$ and checking that their occurrence is indeed independent of $t_1$, distinguishing model variance from the watermark bias.

**Querying strategy**  Recall that our goal is to steer a model into a setting (via an instruction, as we have no access to the completion model) where it makes a choice from some restricted set $\Sigma$. Importantly, we want to ensure enough variability in the model's choices to be able to observe the behavior specific to a Red-Green watermark and perform efficient estimation (in terms of query cost).

Assuming an upper bound $H$ on the context size $h$ of the watermark, we use the following prompt template, parametrized by $t_1$ (an arbitrary length string), $d$ (a single token), and a word list $\Sigma$.

$$\text{f"Complete the sentence \verb|\"|\{t1\} }\underbrace{\{d \cdot H\}}_{t_2}\verb|\"| \text{ using a random word from: } [\{\Sigma\}]."$$

Here, $t_1$ serves as the prefix, and $t_2 = d \cdot H$ as the context. For a watermarked model, changing $t_2$ is the only way to change the red-green vocabulary split, which can greatly affect the model's choice. For unwatermarked models, while we often see bias towards some choices from $\Sigma$, this bias will not strongly depend on $t_2$. This holds for all context sizes $\leq H$ and most aggregation functions, making our setting and the corresponding test directly applicable to different variants of Red-Green schemes.

In our instantiation (illustrated in Fig. 1), we use $N_1$ different $t_1$ of the form *"I bought"*, varying the verb, $N_2$ different values of $d$ from the set of digits, a word list $\Sigma$ of four different plural fruit names, and an example that uses different words to not bias the model towards a specific choice. We empirically observe that this setup minimizes the chance that the model collapses to a single choice or fails to follow the instruction, which often occurred for similar settings based on repetition of words or numbers outside of natural context.

**Estimating the logits**  To collect the data for the test, we query the model in two phases. First, we choose different values of $\Sigma$ until we find one where the model does not always make the same choice, inducing $Q_1$ total queries. We set $x$ to be the most commonly observed word from $\Sigma$. Next, for each $(t_1, t_2)$ pair we query the model until we obtain $K$ valid responses (filtering out failures to follow the instruction), inducing in total $Q_2 \geq N_1 \times N_2$ additional queries.

We use these samples to estimate the model *logits* corresponding to $x$ as $\hat{l}_{t_1,t_2}(x) = \log \frac{\hat{p}_{t_1,t_2}(x)}{1-\hat{p}_{t_1,t_2}(x)}$, where $\hat{p}_{t_1,t_2}(x)$ is the empirically estimated probability of $x$ in the responses. The result of the adversary's querying procedure is a matrix $L_{N_1 \times N_2}$ (visualized in Fig. 1) of such logit estimates.

**Testing watermark presence**  Finally, we describe the statistical test based on the logit estimates $L_{N_1 \times N_2}$. We first estimate $\sigma$, the standard deviation of $\varepsilon'_{t_1,t_2}(x)$, as follows:

$$\hat{\sigma}^2 = \text{median}[\text{Var}_{t_2}(L)], \tag{5}$$

using the empirical median to improve robustness to unpredictable behavior caused by different $t_1$. Then, we calculate the following two binary functions, which flag cases where we believe the model's probability was affected by a watermark:

$$R_x(t_1, t_2) = \mathbb{1}\{\hat{l}_{t_1,t_2}(x) - \text{median}[L] < -r\hat{\sigma}\}, \tag{6}$$

and

$$G_x(t_1, t_2) = \mathbb{1}\{\hat{l}_{t_1,t_2}(x) - \text{median}[L] > r\hat{\sigma}\}, \tag{7}$$

with $r \in \mathbb{R}^+$ a parameter of the test. In practice, to account for model unpredictability, we use the empirical median conditioned on $t_1$ in Eqs. (6) and (7). For simplicity, let us denote $t_1 \in \{1, \ldots, N_1\}$ and $t_2 \in \{1, \ldots, N_2\}$. Let $cnt_x(t_2) = \max\left(\sum_{t_1=1}^{N_1} R_x(t_1, t_2), \sum_{t_1=1}^{N_1} G_x(t_1, t_2)\right)$ count the number of consistently flagged values for fixed $t_2$. We define the following test statistic:

$$S_x(L) = \max_{t_2 \in [N_2]} cnt_x(t_2) - \min_{t_2 \in [N_2]} cnt_x(t_2). \tag{8}$$

The null hypothesis of our test is $\forall t_2 \colon \delta'_{t_2}(x) = 0$, i.e., the model is not watermarked. To obtain a $p$-value, we apply a Monte Carlo permutation test to $S_x$, checking if the flagged values are correlated with $t_2$ in a way that indicates a Red-Green watermark. Namely, we sample a set of permutations $\sigma$ of the matrix $L$ uniformly at random, and calculate a 99% confidence interval of $\Pr[S_x(\sigma(L)) \geq S_x(L)]$, whose upper bound we take as our $p$-value. When this value is small, we interpret that as evidence of a watermark. Because Eq. (3) is permutation invariant when $\delta'_{t_2}(x) = 0$, this ensures that the test does not reject under the null. This completes our method for detection of Red-Green watermarks. In App. B.1, we discuss estimation of $\delta$ and $h$ once the Red-Green watermark presence is confirmed.

## 3 DETECTING FIXED-SAMPLING WATERMARKS

Unlike Red-Green watermarks, the recently proposed *Fixed-Sampling* watermarks do not modify the logit vectors during generation, so estimating the probabilities of model outputs as above is not informative. Instead, the sampling is fully determined by the rotation of the watermark key (Kuditipudi et al., 2024), making the natural vector to exploit when detecting this watermark its *lack of diversity*. Given a prompt for which an unwatermarked model is expected to produce highly diverse outputs, we can use this observation to distinguish between the two, as illustrated in Fig. 1 (middle). For a more formal description of the test we refer to App. C.

As in §2, we start by introducing the needed background. We then formally model the diversity of model outputs, discuss our querying strategy that ensures our assumptions are met, and describe the resulting statistical test. The effectiveness of our method is evaluated in §5.

**Background**    For Fixed-Sampling watermarks, the secret watermark key sequence $\xi$ of length $n_{key}$ is cyclically shifted uniformly at random for each generation to obtain $\bar{\xi}$, and the entry $\bar{\xi}_t$ is used to sample from $l$. In the ITS variant, $\bar{\xi}_t$ is a pair $(u, \pi) \in [0, 1] \times \Pi$, where $\Pi$ defines the space of permutations of the vocabulary $V$. Given the probability distribution $p$ over $V$, obtained by applying the softmax function to $l$, ITS samples the token with the smallest index in the permutation $\pi$ such that the CDF of $p$ with respect to $\pi$ is at least $u$. In the EXP variant, $\bar{\xi}_t$ is a value $u \in [0, 1]$, and we sample the token $\arg\min_{i \in V} -\log(u)/p_i$. The detection, for both variants, is based on testing the correlation between the text and $\xi$ using a permutation test. As noted in §1, the key design goal of ITS and EXP is that, in expectation w.r.t. $\xi$, they do not distort the distribution of the responses. How close to this ideal is achieved in practical implementations is the question we aim to answer.

**Modeling the diversity**    Let $U_n(q, t)$ denote a random variable that counts the number of unique outputs to a fixed prompt $q$ in $n$ queries, each of length exactly $t$ in tokens. We introduce the *rarefaction curve* (visualized in Fig. 1) as

$$R(n) = \mathbb{E}[U_n(q, t)]. \tag{9}$$

For suitable $q$ that enables diversity and large enough $t$, the unwatermarked model exhibits $R(n) = n$. For a Fixed-Sampling watermark (both ITS and EXP variants), the watermark key segment used for sampling is determined by choosing a rotation of the key $\xi$ uniformly at random. As choosing the same rotation for the same prompt and sampling settings will always yield the same output, the number of unique outputs is at most equal to the key size $n_{key}$. The probability that an output $i$ was not produced is given by $(1 - 1/n_{key})^n$. By linearity of expectation, we have the rarefaction curve

$$R(n) = n_{\text{key}} \left( 1 - (1 - \frac{1}{n_{\text{key}}})^n \right). \tag{10}$$

**Querying strategy**    To estimate $R(n)$ of $LM$, we query it with a fixed prompt $q$, using rejection sampling to discard short responses until we obtain a set of $N$ responses of length $t$ tokens (inducing $Q$ total queries). We then repeatedly sample $n$ responses from this set to get a Monte-Carlo estimation of $R(n)$. There are two key considerations. First, we need to ensure that we are in a setting where an unwatermarked model would have $R(n) = n$. To do this, we use the prompt "This is the story of" that reliably causes diverse outputs, and set $t$ high enough to minimize the chance of duplicates. In §5 we experimentally confirm that the number of unique outputs scales exponentially with $t$, and investigate the effect of small temperatures. Second, as larger $n_{key}$ make $R(n)$ closer to linear, we must ensure that $n$ is large enough for our test to be reliable. To do this, we can set an upper bound $\bar{n}_{key}$ on key size, and estimate the number of samples needed for a given power by simulating the test—our experiments show that our test succeeds even on values of $\bar{n}_{key}$ far above practical ones.

**Testing watermark presence**    Finally, to test for presence of a Fixed-Sampling watermark, we use a Mann-Whitney U test to compare the rarefaction curve $R(n) = n$ with the observed rarefaction data obtained as above. If the test rejects the null hypothesis, we interpret this as evidence that the model is watermarked with a Fixed-Sampling scheme. We confirm the effectiveness of our test in §5. In App. B.2 we discuss estimation of the watermark key size $n_{key}$, after the watermark is detected.

## 4 DETECTING CACHE-AUGMENTED WATERMARKS

We proceed to the detection of Cache-Augmented watermarks. While the underlying techniques used in these schemes are often similar to those of the Red-Green and Fixed-Sampling families, we focus on a general approach that exploits the presence of a cache on a fundamental level, and can be generally applied to any Cache-Augmented scheme. Namely, we note that the cache sometimes *reveals the true distribution* of the model, which was also noted in recent work (Pang et al., 2024) as an undesirable property for a watermarking scheme. This implies that the distribution of choices is *cache-conditioned* (as seen in Fig. 1, top), which our adversary will exploit to detect the watermark. We first note that we provide a more formal description of the test in App. C. We also note that Cache-Augmented schemes generally use another underlying mechanism beneath the cache, and may be practical even in a no-cache variant. In particular, for the schemes presented below, the underlying mechanism falls in the Red-Green family, and we experimentally demonstrate in App. F.3 that the no-cache variants of these schemes can be detected by our Red-Green test from §2.

We first recall relevant background, and then discuss our method: we query the model in two phases to probe the two cache-conditioned distributions, and apply a statistical test to detect when they differ, which would indicate the presence of a Cache-Augmented watermark.

**Background** The watermarks that we consider in this section use previous $h$ tokens to reweigh the distribution or apply deterministic sampling at each step. As a key feature motivated by practical distribution preservation, these schemes introduce a *cache* of previously seen contexts. Namely, whenever $y_{t-h:t-1}$ is already present in the cache, they ignore the watermarking procedure and instead fall back to standard generation. We consider three variants: $\delta$-REWEIGHT (Hu et al., 2024), $\gamma$-REWEIGHT (Hu et al., 2024), and DIPMARK (Wu et al., 2024) with parameter $\alpha$ (details in App. D). Previous works do not discuss practical instantiation of the cache. We consider two realistic scenarios: a global cache common to all users or a local per-user cache. With both instantiations, the cache is cleared after a set number $G$ of generations.

**Probing the true distribution** In the first phase of querying, our goal is to find a setting where the distribution of the model under the watermark will differ from its true distribution, and estimate the true distribution. For schemes we focus on, this corresponds to a setting with two choices, where the model is not significantly biased towards any of them. In particular, we use the following prompt:

```
Pick a fruit between: {f₁} and {f₂}. Use the following format: {uc}{f_example},
```

and modify $f_1$, $f_2$, and $f_{example} \neq f_1, f_2$ until we find a setting where the model outputs the two choices roughly uniformly. Crucially, we prefix the prompt with a randomly sampled sufficiently long string of tokens *uc*. As $LM$ will repeat *uc* before providing the answer, this ensures that if a cache is present, after our first query (the result of which we discard) the choice of the model will be made according to the true distribution, as the relevant part of *uc* was cached. Assuming WLOG that $f_1$ is the more likely choice for the model, we query it $Q_1$ times with the same input to obtain $\hat{p_1}$, the estimate of the true probability of the model to pick $f_1$.

**Probing the watermarked distribution** In the second phase, we query $LM$ with the same input, while ensuring that the cache is reset between each query, i.e., the model will respond according to the watermarked distribution. In case of a global cache, it is sufficient to wait for a set amount of time—resetting the cache too infrequently is not a realistic setting for a deployment, as it would on average lead to a weak watermark. The uncommon prefix *uc* ensures that no other user will accidentally insert the same context into the cache. In case of a per-user cache, we can either saturate the cache by asking diverse queries, or use multiple user accounts. We query $LM$ $Q_2$ times and obtain $\hat{p_2}$, the estimate of the probability of $f_1$ under the watermarked distribution.

**Testing watermark presence** For unwatermarked models or those watermarked with a scheme from another family, both of the previous steps were sampling from the same distribution, thus for high enough $Q_1$ and $Q_2$ we expect $\hat{p_1} = \hat{p_2}$. However, for all Cache-Augmented watermarking schemes, these probabilities will differ, indicating that the cache has revealed the true distribution of the model. To test this, we apply a Fischer's exact test with the null hypothesis $\hat{p_1} = \hat{p_2}$. If we observe a low $p$-value, we interpret this as evidence that the model is watermarked with a Cache-Augmented

Table 1: Main results of our watermark detection tests across different models and watermarking schemes. We report median p-values across 100 repetitions of the experiment, and for RED-GREEN schemes additionally over 5 watermarking keys. p-values below 0.05 (test passing) are highlighted in bold. $\delta$R and $\gamma$R denote $\delta$-REWEIGHT and $\gamma$-REWEIGHT schemes, respectively.

| | | Unwatermarked | | Red-Green | | | | Fixed-Sampling | | Cache-Augmented | | |
| | | | | LEFTHASH | | SELFHASH | | ITS/EXP | | DIPMARK/$\gamma$R | | $\delta$R |
| Model | Test | $T=$ 1.0 | $T=$ 0.7 | $\delta,\gamma=$ 2,0.25 | $\delta,\gamma=$ 4,0.5 | $\delta,\gamma=$ 2,0.5 | $\delta,\gamma=$ 4,0.25 | $n_{key}=$ 256 | $n_{key}=$ 2048 | $\alpha=$ 0.3 | $\alpha=$ 0.5 | |
|---|---|---|---|---|---|---|---|---|---|---|---|---|
| MISTRAL 7B | R-G (§2) | 1.000 | 1.000 | **0.000** | **0.000** | **0.000** | **0.000** | 1.000 | 1.000 | 1.000 | 1.000 | 1.000 |
| | Fixed (§3) | 0.938 | 0.938 | 0.938 | 0.938 | 0.938 | 0.938 | **3.7e-105** | **1.8e-06** | 0.938 | 0.938 | 0.938 |
| | Cache (§4) | 0.570 | 0.667 | 0.607 | 0.608 | 1.000 | 0.742 | 0.638 | 0.687 | **2.4e-4** | **2.1e-3** | **5.6e-27** |
| LLAMA2 13B | R-G (§2) | 0.149 | 0.663 | **0.000** | **0.000** | **0.000** | **0.000** | 0.121 | 0.128 | 0.149 | 0.149 | 0.149 |
| | Fixed (§3) | 0.972 | 0.869 | 0.938 | 0.938 | 0.938 | 0.938 | **8.1e-122** | **1.5e-07** | 0.938 | 0.938 | 0.938 |
| | Cache (§4) | 0.708 | 0.573 | 0.511 | 0.807 | 0.619 | 0.710 | 0.518 | 0.692 | **1.8e-2** | **5.3e-3** | **6.7e-32** |
| LLAMA2 70B | R-G (§2) | 1.000 | 1.000 | **0.000** | **0.020** | **0.020** | **0.000** | 1.000 | 1.000 | 1.000 | 1.000 | 1.000 |
| | Fixed (§3) | 0.938 | 0.525 | 0.968 | 0.968 | 0.987 | 0.975 | **4.5e-125** | **1.7e-08** | 0.938 | 0.968 | 0.938 |
| | Cache (§4) | 0.596 | 0.620 | 0.657 | 0.639 | 0.651 | 0.608 | 0.463 | 0.818 | **1.5e-3** | **4.4e-3** | **5.8e-28** |

scheme. Our experiments in §5 demonstrate that this test is robust to different scheme variants, and does not lead to false positives when $LM$ is unwatermarked or uses a scheme from another family. In App. B.3 we discuss how to distinguish different variants of Cache-Augmented schemes.

## 5 EXPERIMENTAL EVALUATION

In this section, we apply the tests introduced in §2–§4 to a wide range of models and watermarking schemes, and confirm their effectiveness in detecting watermarks. In §5.1 we show that the adversary can reliably detect the watermarking scheme used (if any) at a low cost, across a wide range of practical settings (extended results provided in App. F.1). In §5.2, we provide further experimental insights into the tests' robustness. Finally, in §5.3, we demonstrate that our tests are practical and can be applied to real-world LLM deployments.

A study of how the power of our tests scales with the number of queries is deferred to App. E. In App. F.2–F.6 we present additional experiments, where we demonstrate the robustness of our tests to more schemes and scheme variants, including adversarial modifications and attempts to make schemes undetectable.

In particular, we study the multi-key variant of Red-Green schemes (App. F.2), the no-cache variant of Cache-Augmented schemes (App. F.3), and the SynthID-Text (Google DeepMind, 2024) scheme (App. F.4), released after the first version of our work was completed. We further study an adversarial modification which selectively disables the watermark (App. F.5), and a new entropy-conditioned variant of the AAR watermark (Aaronson, 2023) that is inspired by Christ et al. (2024) (App. F.6).

### 5.1 MAIN EXPERIMENTS: DETECTING WATERMARKING SCHEMES

We perform our main set of experiments to verify the effectiveness of the tests introduced in §2–§4 in detecting watermarking schemes in realistic scenarios.

**Experimental setup**  We run all our tests on seven different instruction fine-tuned models (MISTRAL-7B, LLAMA2-7B, -13B, and -70B, LLAMA3-8B, YI1.5-9B and QWEN2-7B), in eleven different scenarios. We here present results of a subset of those, and defer the rest as well as additional metrics to App. F.1. In each scenario, each model is either unwatermarked (where we vary the temperature) or watermarked with a certain scheme from the three main families (where we vary the particular scheme and its parameters). If our tests are reliable, we expect to see low p-values only when the model is watermarked exactly with the scheme family that we are testing for.

For Red-Green tests, we set $N_1 = 10, N_1 = 9, r = 1.96$, a different $\Sigma$ per model based on the first $Q1$ samples, use 100 samples to estimate the probabilities, and use 10000 permutations in the test.

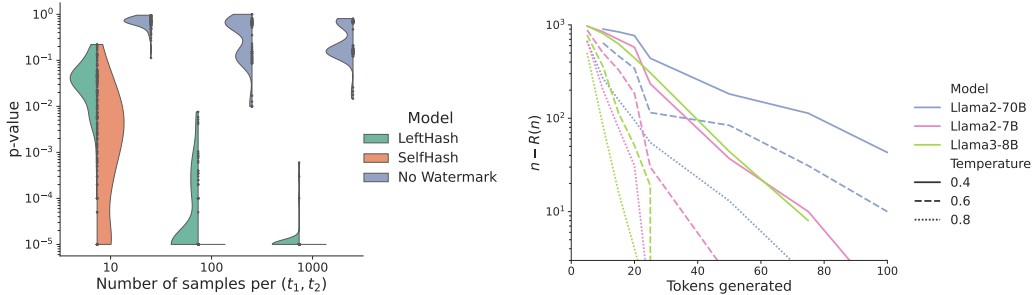

Figure 2: *Left*: distribution of bootstrapped p-values of the Red-Green test on LLAMA2-13B with $(\delta, \gamma) = (2, 0.25)$, for different sample sizes. We see reliable results for 100 or more samples. *Right*: the diversity gap $n - R(n)$ on log scale in different settings. Linear behavior means that diversity scales exponentially with $t$, and we see that the assumption of $R(n) = n$ can be easily met in practice.

We randomly sample a set of 5 watermark keys and execute 5 independent repetitions of the entire experiment, each time using exactly 1 of those 5 keys. For Fixed-Sampling tests, we use $n = 1000$ queries and set $t = 50$. For Cache-Augmented tests, we use $Q1 = Q2 = 75$ and assume the cache is cleared between queries in the second phase.

**Results: reliable watermark detection**  Our main results are shown in Table 1, where we report the median p-values, as in (Kuditipudi et al., 2024), for each (model, test, scenario) tuple across 100 repetitions of each experiment. Across all experiments, all three tests reject the null hypothesis (*the specific watermark is not present*) at a 95% confidence level only when the scheme from the target family is indeed applied to the model. This confirms that our tests are reliable in detecting watermarks, and robust with respect to the model and the specific parameters of the scheme, emphasizing our tests' generalization to all schemes based on the same foundational principles. In particular, our Red-Green tests are robust to the seeding scheme, the logit bias $\delta$, and the green token fraction $\gamma$; our Fixed-Sampling tests maintain high confidence even for high values of $n_{key}$ and those higher than the number of queries; finally, our Cache-Augmented tests are robust to all details of the underlying scheme. Our results also suggest that unrelated model modifications do not cause false positives:, as no test passes when the model is unwatermarked or watermarked with a different scheme family.

Our tests do not incur significant costs for the adversary, making them easily applicable in practice. While the cost of a particular run varies across models and scenarios, we estimate the average cost of the tests to be below \$3 for Red-Green, \$0.3 for Fixed-Sampling, and \$0.1 for Cache-Augmented tests, assuming latest OpenAI GPT4O pricing (see App. G for a detailed estimation of the costs).

## 5.2 ADDITIONAL EXPERIMENTS: VALIDATING THE ASSUMPTIONS

We present two additional experiments to validate the assumptions made in our tests and provide further insight into their behavior in practical settings.

**Sampling in Red-Green tests**  The Red-Green test (§2) relies on the sampling of model outputs to estimate the probabilities of the model. As the resulting p-value is computed assuming knowledge of true probabilities, this raises the question of the impact of the sampling error on our results. To heuristically mitigate this, we propose a bootstrapping procedure, where for fixed $(t_1, t_2)$, we sample with replacement from a single set of model outputs, and report the median p-value $p_{med}$ across such samples. In Fig. 2 (Left) we report the resulting distribution of $p_{med}$, where one point corresponds to one independent run. We see that already for 100 samples per $(t_1, t_2)$ (as used in Table 1), the p-value distribution is narrow and the false positive rate due to sampling error is well controlled. This confirms that our test is robust against sampling error and can be used in realistic settings, without additional access to model logprobs. For computational reasons, we did not apply this correction in Table 1, where we still observe reliable results in the median case across experiment repetitions.

**Diversity assumption in Fixed-Sampling tests**  As detailed in §3, the Fixed-Sampling test relies on the unwatermarked model being sufficiently diverse, i.e., for the number of unique outputs

Table 2: The results of our watermark detection tests on black-box LLM deployments.

|            | GPT4  | CLAUDE 3 | GEMINI 1.0 PRO |
|------------|-------|----------|----------------|
| R-G (§2)   | 0.998 | 0.638    | 0.683          |
| Fixed (§3) | 0.938 | 0.844    | 0.938          |
| Cache (§4) | 0.51  | 0.135    | 0.478          |

$R(n) = \mathbb{E}[U_n(q, t)]$ after $n$ queries with prompt $q$ and response length $t$, it should hold that $R(n) = n$. Our goal is to show that we can easily choose $t$ such that this property holds across different settings.

To this end, we hypothesize that the number of unique outputs converges to $n$ exponentially fast as the response length $t$ increases. In particular, we assume

$$R(n) = n - \lfloor n \cdot \exp(-\alpha(T)t) \rfloor, \tag{11}$$

where $\alpha(T)$ is a monotonic function of the temperature $T$. To verify this hypothesis, we measure $n - R(n)$ on several models and temperatures, and show the result on log scale in Fig. 2 (Right). If Eq. (11) holds, we expect the resulting relationship to be linear, which is indeed confirmed by our results. While $\alpha$ (the slope of the line) varies across settings, we see that a bit over 200 tokens would be sufficient for the line to drop to 0 (not visible on the log scale plot). This holds even in cases impractical for deployment of Fixed-Sampling watermarks such as $T = 0.4$ (Kuditipudi et al., 2024; Piet et al., 2023), indicating that $R(n) = n$ and our assumption is met, validating our p-values.

### 5.3 DETECTING WATERMARKS IN DEPLOYED MODELS

Finally, we demonstrate the applicability of the statistical tests introduced in §2–§4, by applying them to popular black-box LLM deployments: GPT4, CLAUDE 3, and GEMINI 1.0 PRO. We use the same experimental setup as in §5.1, and use the API access for efficiency reasons—we do not rely on any additional capabilities, and our tests could be easily run in the web interface. For the Cache-Augmented tests, we assume a global cache that clears after 1000 seconds. For the Fixed-Sampling test on CLAUDE 3, due to its lack of diversity, we used $t = 75$ tokens per query to ensure the hypotheses are met. Our results in Table 2 show that the null hypothesis is not rejected for any of the models and any of the tests. Hence, we can not conclude on the presence of a watermark.

## 6 RELATED WORK

**Language model watermarking** Besides the approaches by (Hu et al., 2024; Kirchenbauer et al., 2023; Kuditipudi et al., 2024) introduced above, there are various methods building on similar ideas. Hou et al. (2024); Liu et al. (2024b); Ren et al. (2024) all apply variations of (Kirchenbauer et al., 2023) on semantic information, while Gu et al. (2024) distills a new model from the output of a watermarked model. Similarly, Liu et al. (2024a) apply a Red-Green scheme using a learned classifier instead of hash functions. A range of works on multi-bit watermarking (Wang et al., 2024; Yoo et al., 024s) aim to not only watermark generated texts but encode additional information in the watermark.

**Attacks on language model watermarking** Attacks on LLM watermarks have so far been mainly investigated in terms of scrubbing (Jovanović et al., 2024; Kirchenbauer et al., 2023; Sadasivan et al., 2023) (i.e., removal of a watermark) and spoofing (Gu et al., 2024; Jovanović et al., 2024; Sadasivan et al., 2023) (i.e., applying a watermark without knowing $\xi$). Notably, Jovanović et al. (2024) showed that observing watermarked texts can facilitate both attacks on various distribution-modifying schemes, disproving common robustness assumptions (Kirchenbauer et al., 2024). However, using this and similar attacks as means of practical watermark detection is infeasible, as they generally offer no way to quantify the attack's success—in contrast, we aim to provide rigorous statements about scheme presence. Further, such attacks incur significantly higher query costs than necessary for detection (as our work demonstrates), and in some cases assume certain knowledge of the watermark parameters, a setting fundamentally at odds with our threat model of black-box watermark detection.

The closest related work to ours is Tang et al. (2023), that tackles the problem of watermark detection in strictly simpler settings where the adversary either has access to an unwatermarked counterpart

of the target model, or can access full model logits. Such knowledge is commonly not available in practice, limiting the applicability of this approach. To the best of our knowledge, no work has developed methods for detecting the presence of a watermark in a realistic black-box setting.

**Extracting data from black-box models**   With many of the most potent LLMs being deployed behind restricted APIs, the extraction of model details has been an active area of research. This includes, e.g., the reconstruction of a black-box model tokenizer (Rando and Tramèr, 2024) as well as the extraction of the hidden-dimensionality or the weights of the embedding projection layer (Carlini et al., 2024). Naseh et al. (2023) have shown practical attacks to recover the decoding mechanism of non-watermarked black-box models. Given access to output logits, Li and Fung (2013) have further demonstrated that it is possible to train an inversion model that aims to recover the input prompt.

## 7   CONCLUSION AND LIMITATIONS

In this paper, we have focused on the problem of detecting watermarks in large language models (LLMs) given only black-box access. We developed rigorous statistical tests for the three most prominent scheme families, and validated their effectiveness on a wide range of schemes and real-world models. Our results show that most popular practical watermarking schemes are detectable.

**Limitations**   One limitation of our tests is that they are restricted to the three scheme families discussed in §2–§4. These are however the most prominent in the literature, and as our tests are based on fundamental properties of these scheme families, they should generalize to more variants and combinations of the underlying ideas. We confirm this in our additional experiments on a multi-key variant of Red-Green schemes (App. F.2), a no-cache variant of Cache-Augmented schemes (App. F.3), and SynthID-Text (Dathathri et al., 2024) (App. F.4). As shown in App. F.5 our tests are also robust to adversarial scheme modifications, aimed at bypassing our tests.

It is still possible that a model provider deploys a practical scheme based on an entirely novel idea, which our tests would not be able to detect. A promising starting point may be the scheme proposed by Christ et al. (2024), which has theoretical undetectability guarantees, but is not yet practically viable. We discuss this in more detail in App. F.6, and introduce and evaluate a proof-of-concept variant of AAR (Aaronson, 2023) that incorporates similar ideas. Our preliminary results show that this approach indeed enhances stealth, but at the cost of reducing the watermark strength and potentially other important properties, which we do not evaluate in this work. We hope our initial investigation inspires more thorough research in this direction.

As another limitation, while we base our tests on fundamental properties of scheme families, we have no theoretical guarantees of power and only provide empirical evidence based on the tested schemes.

Finally, our tests make several model assumptions, such as symmetric error terms, perfect sampling, and the unlikely event of the red-green split (in Red-Green schemes) being the same for all contexts on the observed domain. While we validate that these assumptions are sufficiently met on several open-source models, we cannot guarantee that all models adhere to them.

## ACKNOWLEDGEMENTS

The work has received funding from the Swiss State Secretariat for Education, Research and Innovation (SERI).

## ETHICS AND REPRODUCIBILITY STATEMENTS

Our work may be used by malicious actors to detect watermarks on deployed models. This could allow them to favor unwatermarked models or help them mount additional attacks that could negatively affect the model provider that deployed the watermark. Yet, we believe the benefits of raising awareness about the easy detectability of most watermark schemes outweigh the risks of our work.

To ensure reproducibility, before each experiment in §5 and App. B, E and F, we thoroughly describe the experimental setup to ensure reproducibility. The code needed to reproduce our tests

on open-source models for the Red-Green watermarking scheme family (§2), the Fixed-Sampling family (§3), and the Cache-Augmented family (§4) is available at `https://github.com/eth-sri/watermark-detection`. This also includes the code used to perform the tests on deployed models (§5.3), which shows a concise way to perform the test in practical settings, and the additional experiments in App. F.

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

# A   PRACTICAL IMPLICATIONS OF WATERMARK DETECTION

In this section, we discuss some practical use cases for detecting whether a given black-box API is watermarked, as ways to provide additional motivation for our work.

The objective behind watermarking an LLM is to enable the detection of whether a given text was generated by a specific LLM. In practice, it should allow both holding a model provider accountable for harmful text generated by its model and holding users accountable for using an LLM in scenarios where its use is inappropriate or forbidden.

Being able to detect a watermark behind an LLM deployment provides a malicious user with multiple opportunities. First, detection is a common prerequisite for performing spoofing attacks (Gu et al., 2024; Jovanović et al., 2024; Zhang et al., 2024), where a malicious user learns the watermark in order to generate arbitrary watermarked text without using the watermarked model. Such attacks can be used to discredit a model provider by generating text that appears to be genuinely watermarked and attributing it to the model provider. Second, detection is a prerequisite for assisted scrubbing attacks (as in Jovanović et al. (2024)), where a malicious user can more successfully remove the watermark from an LLM generated text compared to blindly rewriting the watermarked texts. Consequently, such malicious users can nullify any positive effects associated with the watermark deployment. Lastly, knowing that a particular LLM is watermarked may lead a malicious user to avoid using that LLM entirely and instead favor another LLM that is not known to be watermarked.

Hence, knowing how detectable schemes are in practice is, besides theoretical interest, important for model providers or legal authorities to have realistic expectations regarding the effectiveness and pitfalls of a given watermarking scheme

# B   ESTIMATING SCHEME PARAMETERS

In this section, we describe how, mostly leveraging the queries already performed during the detection tests, we can estimate the main parameters of the detected watermarking scheme. This demonstrates the soundness of our modeling assumptions in §2–§4 from which we derive all the estimators.

## B.1   ESTIMATION OF RED-GREEN WATERMARKING SCHEME PARAMETERS

If the null hypothesis (*the model not being Red-Green watermarked*) is rejected, we can then estimate the scheme-specific watermark parameters $\delta$ and the context size $h$ using mostly the same data as used in the test. First, we describe the estimators for both parameters, and then discuss their practicality by analyzing their performance on multiple models.

**Description of estimators**   To estimate $\delta$, we establish a parametrized model based on Eq. (2) that relies on our flagging function from Eq. (7), and additional estimates $\hat{l}_{t_1,t_2}(w)$ for every $w \in \Sigma$, computed on the same data as above, requiring no additional queries. For each $w \in \Sigma$, we set:

$$\hat{l}_{t_1,t_2}(w) = \bar{w}_{t_1} + G_w(t_1,t_2)\bar{\delta} - \log\left(\sum_{w' \in \Sigma \setminus \{w\}} \exp\left(\bar{w}'_{t_1} + G_{w'}(t_1,t_2)\bar{\delta}\right)\right), \qquad (12)$$

and set $\forall t_1$, $\bar{w}_{t_1} = 0$ for a single $w \in \Sigma$, as logits are shift-invariant. Fitting the parameters $\bar{\delta}$ and all $\bar{w}_{t_1}$ by minimizing the mean squared error with gradient descent allows us to recover $\delta/T$ as $\bar{\delta}$. If $T$ is known or estimated separately, this term can be removed.

Let $h \in \mathbb{N}$ denote the context size, i.e., the number of *previous* tokens considered by the watermarking scheme. To estimate $h$, we use the same prompt template as in §2, with a fixed prefix $t_1$ and digit $d$, but with a varying $H \in \mathbb{N}$ and perturbation digit $d'$ prepended to $d$.

```
f"Complete the sentence \"{t1} {d'}{d · H}\" using a random word from: [{Σ}]."
```

---

**Algorithm 1** Context Size Estimation

---

**Require:** Two tokens $t_1, d$, set of perturbation tokens $\Sigma'$, set of tokens $\Sigma$, number of samples $N$, upper bound on the context size $H_{max}$, $x^* \in \Sigma$, $\rho \in [0, 1]$, LM $\mathcal{M}$

1: $\hat{p} \leftarrow \mathbf{0}, \hat{p} \in [0, 1]^{1 \times 1 \times |\Sigma'| \times H_{max} \times |\Sigma|}$
2: **for** each token $d'$ in $\Sigma'$ **do**                      ▷ Generate the estimator data
3:      **for** $H$ from 1 to $H_{max}$ **do**
4:          prompt $\leftarrow$ f"Complete the sentence $\{t_1\}$ $\{d'\}$ $\{d.H\}$ using a word in $\{\Sigma\}$"
5:          **for** $i$ from 1 to $N$ **do**
6:              resp $\leftarrow \mathcal{M}(\text{prompt})$
7:              $x_i \leftarrow \text{ParseResponse}(\text{resp})$
8:              $\hat{p}_{t_1,d,d',H}(x_i) \leftarrow \hat{p}_{t_1,d,d',H}(x_i) + 1/N$
9:          **end for**
10:      **end for**
11: **end for**
12: **for** each token $d'$ in $\Sigma'$ **do**                      ▷ Build logit estimation matrix
13:      **for** $H$ from 1 to $H_{max}$ **do**
14:          **for** $i$ from 1 to $N$ **do**
15:              $\hat{l}_{t_1,d,d',H}(x) \leftarrow \log \frac{\hat{p}_{t_1,d,d',H}(x)}{1-\hat{p}_{t_1,d,d',H}(x)}$
16:          **end for**
17:      **end for**
18: **end for**
19: $\hat{h} \leftarrow 1$
20: **for** $H$ from 2 to $H_{max}$ **do**
21:      $p \leftarrow \text{MoodTest}_{D' \sim \Sigma'}(\hat{l}_{t_1,d,D',H}(x^*), \hat{l}_{t_1,d,D',H-1}(x^*))$
22:      **if** $p \leq \rho$ **then**
23:          $\hat{h} \leftarrow H$
24:          **break**
25:      **end if**
26: **end for**
27: **return** $\hat{h}$

---

The probability distribution of the model output will abruptly change when $H$ exceeds the context size $h$, as the change in $d'$ will not alter the red/green split of the vocabulary. We then compute the corresponding logit estimator $\hat{l}_{t_1,d,d',H}$ for every variation of the above prompt by sampling $N$ times from the model. For a Red-Green watermark with context size $h$, we then have the following property:

$$\forall t_1, d \in V, \forall h_1 < h, h_2 < h, \forall d' \in V, \hat{l}_{t_1,d,d',h_1} \approx \hat{l}_{t_1,d,d',h_2} \tag{13}$$

$$\forall t_1, d \in V, \forall h_1 \geq h, h_2 \geq h, \forall d' \in V, \hat{l}_{t_1,d,d',h_1} \approx \hat{l}_{t_1,d,d',h_2} \tag{14}$$

$$\forall t_1, d \in V, \forall h_1 < h, h_2 \geq h, \forall d' \in V, \hat{l}_{t_1,d,d',h_1} \neq \hat{l}_{t_1,d,d',h_2} \tag{15}$$

Therefore, for a given pair $(d, t_1) \in \Sigma \times \Sigma$, we test the difference between the distributions of $\hat{l}_{t_1,d,d',H-1}$ and $\hat{l}_{t_1,d,d',H}$ using a Mood test. Here, $d'$ is treated as a random variable uniformly sampled from a subset of $\Sigma$. Starting with $H = 1$ and incrementing $H$ by 1 at each step, we continue this process until the Mood test exceeds a given threshold $\rho \in [0, 1]$. The value of $H$ at which the threshold is first rejected is set to $\hat{h}_{t_1,d}$. The estimator computed using a fixed $(d, t_1) \in \Sigma \times \Sigma$ is detailed in Algorithm 1.

Estimating $\gamma$ is more challenging, as in contrast to $\delta$, this parameter is not directly reflected in the logits but rather defines a more global behavior of the scheme. This is particularly difficult for schemes with self-seeding, as the rejection sampling interferes with the behavior of $\gamma$. We leave further exploration of this problem to future work.

**Experimental results** We computed the estimator for $\delta$ on the LeftHash variant with $\gamma = 0.25$ and varying $\delta$ from 0 to 4. The results are shown in Fig. 3, with the 95% confidence intervals reflecting the sampling error. The estimator successfully estimates $\delta$ for all models with a sufficient number of

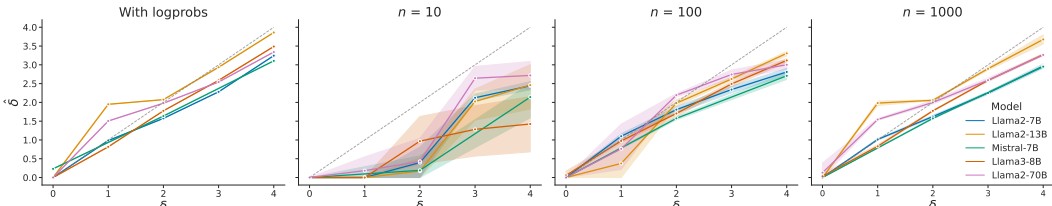

Figure 3: Estimation of $\delta$ for different models using LeftHash with $\gamma = 0.25$. The number of samples used increases from left to right, with the leftmost plot assuming direct access to the log-probabilities. The estimation is done on the same data as the test. Error bars are given by the 95% bootstrapped confidence interval with respect to the sampling of the model outputs.

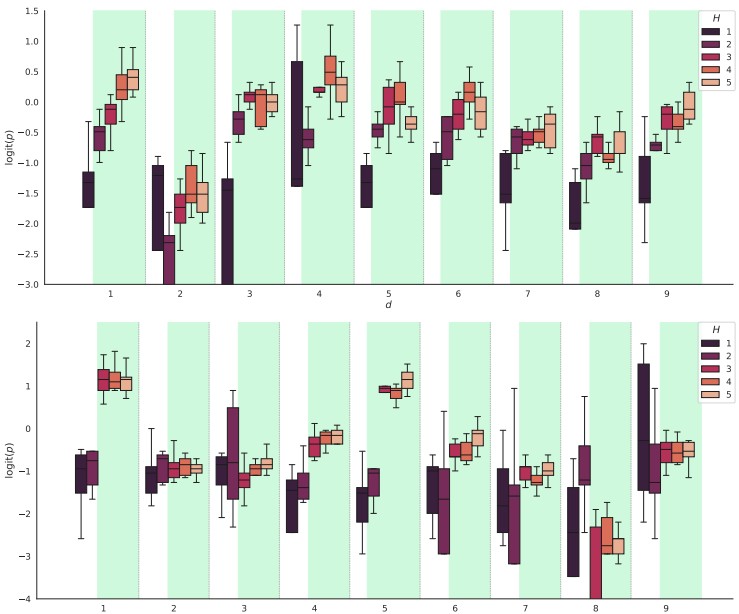

Figure 4: Estimation of the context size $h$ in Red-Green watermarks with LeftHash $h = 2$ (top) and LeftHash $h = 3$ (bottom) on LLAMA2-7B. Each box corresponds to the distribution of $\hat{l}_{t_1,d,d',H}$. The green shading corresponds to the region where $\hat{h}_{t_1,d} \geq H$. A fixed $t_1$ is used across all plots.

Table 3: Key length estimation for Fixed-Sampling watermarks using non-linear regression on the rarefaction curve.

| Key length | LLAMA2-7B | LLAMA2-13B | LLAMA2-70B | LLAMA3-8B | MISTRAL-7B |
|---|---|---|---|---|---|
| 256 | $259 \pm 0.6$ | $259 \pm 0.5$ | $256 \pm 0.5$ | $257 \pm 0.5$ | $256 \pm 0.6$ |
| 2048 | $1978 \pm 10$ | $2107 \pm 12$ | $2006 \pm 13$ | $2070 \pm 14$ | $1831 \pm 10$ |

samples, using only the estimated output probabilities of the model. It is also consistent across all models tested, suggesting that the model assumptions in Eq. (1) are met in practice.

Estimating the context size for Red-Green schemes requires performing a new attack once the model is flagged as watermarked. We estimate the context size for three different models (MISTRAL-7B, LLAMA2-13B and LLAMA2-70B) using LeftHash with $\delta = 2$ and $\gamma = 0.25$. The estimation process requires an additional 5,000 queries, and the estimator successfully determines the context size for all models. However, the estimator is less robust on the SelfHash variant due to the self-seeding algorithm, which leads to higher probabilities for tokens in $\Sigma$ being in the green vocabulary, and thus diminishes the perturbation's significance and resulting in false negatives in Mood's test. Therefore, the procedure stated above produces a lower bound for the context size. To mitigate this issue, we use the estimator across 10 different $t_2$ and then consider the median of the 10 estimators as our final estimator. This estimator applied on the SelfHash variant with $\delta = 2$ and $\gamma = 0.25$ is successful on all three models. It also does not change the results on LeftHash and can be used as a more robust estimator for $h$ in all cases, when the additional cost of $50,000$ queries is not prohibitive. In Fig. 4, we see that for both LeftHash $h = 2$ and LeftHash $h = 3$, the distribution of each logit estimator $\hat{l}_{t_1,d,d',H}$ with respect to $d'$ changes significantly when $H$ exceeds $h$. The $H$ at which this abrupt change in distribution occurs corresponds to the estimator $\hat{h}$ and matches the actual value of $h$.

### B.2 ESTIMATION OF FIXED-SAMPLING WATERMARKING SCHEME PARAMETERS

Our approach does not distinguish between the variant of the Fixed-Sampling watermark used (ITS or EXP), as the diversity property that we exploit is common to both. The only other relevant parameter of Fixed-Sampling schemes is $n_{\text{key}}$. To estimate it, we use non-linear regression on the rarefaction curve using (10) and the same data that we used for the presence test, and compute the confidence intervals using bootstrapping.

Our results are given in Table 3. We see that the estimator is consistent across different models and remains relatively precise even for values of $n_{\text{key}}$ higher than the number of queries.

### B.3 ESTIMATION OF CACHE-AUGMENTED WATERMARKING SCHEME PARAMETERS

For Cache-Augmented watermarks, we can estimate which scheme variant is present, and if the variant is DIPMARK, attempt to learn the value of $\alpha$ (recall that $\alpha = 0.5$ corresponds to $\gamma$-REWEIGHT). To do this, we use the same approach as in §4 to obtain $\hat{p_1}$ and $\hat{p_2}$, where WLOG we assumed $p_1 > 0.5$. If we observe $\hat{p_2} = 0$ this directly implies a $\delta$-REWEIGHT watermark. If we observe $\hat{p_2} \in (0, 1)$, we learn the following: if $\hat{p_2} = 2\hat{p_1} - 1$ then $\alpha > 1 - \hat{p_1}$, otherwise $\alpha = |\hat{p_1} - \hat{p_2}|$. The bound in the first case can be further improved with additional queries with different $p_1$. Finally, if we observe $\hat{p_2} = 1$ we repeat the whole procedure in total $K$ times, following the same case distinction—if $\hat{p_2} = 1$ repeats in all $K$ runs, we conclude that the model is watermarked with $\delta$-REWEIGHT.

Using the same parameters as the one for the test, we distinguish with 100% accuracy between a DIPMARK and a $\delta$-REWEIGHT watermark. However, the estimation of $\alpha$ becomes unreliable for higher values of $\alpha$, especially for smaller models. One of the reasons for this are the failures of the model to follow the instruction, that are more common in the presence of the uncommon prefix $uc$. While the detection test in §4 was robust to such behavior, this does not hold for the estimation of $\alpha$.

## C ALGORITHMIC DESCRIPTIONS OF THE DETECTION TESTS

We present an additional algorithmic description of the Red-Green test (§2) in Algorithm 2, the Fixed-Sampling test (§3) in Algorithm 4 and the Cache-Augmented test (§4) in Algorithm 5.

---

**Algorithm 2** Red-Green Test

---

**Require:** Token set $\Sigma$, token set $T_1$, token set $T_2$, context size $H$, number of samples $N$, number of permutation $M$, $x^* \in \Sigma$, $r > 0$, LM $\mathcal{M}$
1: $\hat{p} \leftarrow \mathbf{0}, \hat{p} \in [0,1]^{|T_1| \times |T_2| \times |\Sigma|}$
2: **for** each token $t_1$ in $T_1$ **do**              $\triangleright$ Generate test data
3:      **for** each token $d$ in $T_2$ **do**
4:          $t_2 \leftarrow d \cdot H$              $\triangleright$ Concatenation
5:          prompt $\leftarrow$ f"Complete the sentence {t1} {t2} using a random word from {$\Sigma$}"
6:          **for** $i$ from 1 to $N$ **do**
7:              resp $\leftarrow \mathcal{M}$(prompt)
8:              $x_i \leftarrow$ ParseResponse(resp)
9:              $\hat{p}_{t_1,t_2}(x_i) \leftarrow \hat{p}_{t_1,t_2}(x_i) + 1/N$
10:          **end for**
11:      **end for**
12: **end for**
13: **for** each token $t_1$ in $T_1$ **do**              $\triangleright$ Build logit estimation matrix
14:      **for** each token $d$ in $T_2$ **do**
15:          $t_2 \leftarrow d \cdot H$              $\triangleright$ Concatenation
16:          $\hat{l}_{t_1,t_2}(x) \leftarrow \log \frac{\hat{p}_{t_1,t_2}(x)}{1-\hat{p}_{t_1,t_2}(x)}$
17:      **end for**
18: **end for**
19: $L \leftarrow \hat{l}_{t_1,t_2}(x^*)$
20: $S \leftarrow$ Statistic($L$)
21: $P \leftarrow 0$              $\triangleright$ Permutation test
22: **for** each $i$ in range $M$ **do**
23:      $\sigma \in \mathcal{U}(S_{|T_1| \times |T_2|})$
24:      $P \leftarrow P + [\text{Statistic}(\sigma(L)) \geq S]$          $\triangleright$ $L$ is permuted element wise
25: **end for**
26: $p \leftarrow P/M$
27: **return** $p$

---

**Algorithm 3** Red-Green Statistic

---

**Require:** Logit estimation matrix $L$, $r > 0$, token list $T_1$, token list $T_2$
1: $R \leftarrow \mathbf{0}, R \in \mathbb{N}^{|T_1| \times |T_2|}$
2: $G \leftarrow \mathbf{0}, G \in \mathbb{N}^{|T_1| \times |T_2|}$
3: $\hat{\sigma}^2 \leftarrow \text{median}\left[\text{Var}_{T_2}(L)\right]$
4: $m \leftarrow \text{median}\left[L\right]$
5: **for** $t_1$ in $T_1$ **do**
6:      **for** $t_2$ in $T_2$ **do**
7:          **if** $L[t_1, t_2] - m < -r\hat{\sigma}^2$ **then**
8:              $R[t_1, t_2] = 1$          $\triangleright$ Abnormally low logits are considered Red
9:          **end if**
10:          **if** $L[t_1, t_2] - m > r\hat{\sigma}^2$ **then**
11:              $G[t_1, t_2] = 1$          $\triangleright$ Abnormally high logits are considered Green
12:          **end if**
13:      **end for**
14: **end for**
15: count $\leftarrow \mathbf{0}$, count $\in \mathbb{N}^{|T_2|}$          $\triangleright$ Count the number of Red/Green tokens per fixed context
16: **for** $t_2$ in $T_2$ **do**
17:      count$[t_2] \leftarrow \max\left(\sum_{t_1 \in T_1} R[t_1, t_2], \sum_{t_1 \in T_1} G[t_1, t_2]\right)$
18: **end for**
19: $S \leftarrow \max_{t_2 \in T_2} \text{count}[t_2] - \min_{t_2 \in T_2} \text{count}[t_2]$
20: **return** $S$

---

---

**Algorithm 4** Fixed-Sampling Test

---

**Require:** Number of queries $Q$, number of tokens to generate $t$, prompt, LM $\mathcal{M}$
1: $R \leftarrow \mathbf{0}, R \in \mathbb{N}^Q$
2: seen $\leftarrow \emptyset$
3: **for** $i$ from 1 to $Q$ **do**
4:      $x_i \leftarrow \mathcal{M}(\text{prompt})$
5:      **if** $x_i \notin$ seen **then**
6:          $R[i] \leftarrow R[i-1] + 1$
7:          seen $\leftarrow$ seen $\cup \{x_i\}$
8:      **else**
9:          $R[i] \leftarrow R[i-1]$
10:      **end if**
11: **end for**
12: $p \leftarrow$ Mann-Whitney$(R, [1, 2, \ldots, Q])$
13: **return** $p$

---

**Algorithm 5** Cache-Augmented Test

---

**Require:** Number of queries $Q_1$, Number of queries $Q_2$, $f_1 \neq f_2 \in V$, $uc \in V^*$, LM $\mathcal{M}$
1: prompt $\leftarrow$ f"Pick randomly $\{f_1\}$ of $\{f_2\}$ and prepend $\{uc\}$ to your choice."
2: $\hat{p}_1 \leftarrow 0$                                       ▷ **Phase 1:** Probing the watermarked distribution
3: **for** $i$ from 1 to $Q_1$ **do**
4:      resp $\leftarrow \mathcal{M}(\text{prompt})$
5:      $x_i \leftarrow$ ParseResponse(resp)
6:      **if** $x_i = f_1$ **then**
7:          $\hat{p}_1 \leftarrow \hat{p}_1 + 1$
8:      **end if**
9: **end for**
10: $\hat{p}_2 \leftarrow 0$                                   ▷ **Phase 2:** Probing the watermarked distribution
11: **for** $i$ from 1 to $Q_2$ **do**
12:      Clear the cache
13:      resp $\leftarrow \mathcal{M}(\text{prompt})$
14:      $x_i \leftarrow$ ParseResponse(resp)
15:      **if** $x_i = f_1$ **then**
16:          $\hat{p}_2 \leftarrow \hat{p}_2 + 1$
17:      **end if**
18: **end for**
19: $p \leftarrow$ FischerExact$((\hat{p}_1, Q_1 - \hat{p}_1), (\hat{p}_2, Q_2 - \hat{p}_2))$
20: **return** $p$

---

# D    DETAILS OF CACHE-AUGMENTED SCHEMES

We provide the details of the three Cache-Augmented watermarking schemes considered in this work: $\delta$-REWEIGHT (Hu et al., 2024), $\gamma$-REWEIGHT (Hu et al., 2024), and DIPMARK (Wu et al., 2024), that were omitted from §4.

All three variants, at each generation step $t$, jointly hash the watermark key $\xi \in \mathbb{Z}_2^K$ and the preceding context $y_{t-h:t-1}$ (commonly setting $h = 5$) using SHA256, and use the result as a seed to sample a *code* $E_t \in P_E$ uniformly at random. Let $p$ denote the probability distribution over $V$, obtained by applying the softmax function to the logits. For the $\delta$-REWEIGHT variant, $P_E = [0, 1]$, and the code $E_t$ is used to sample the token in $V$ with the smallest index, such that the CDF of $p$ is at least $E_t$. For the $\gamma$-REWEIGHT variant and DIPMARK, $P_E$ is the space of permutations of $V$. For $\gamma$-REWEIGHT, we transform $p$ to a new distribution $p'$ by, for each token $i \in V$, setting $p'(i) = f_2(f_1(E_t(i))) - f_2(f_1(E_t(i) - 1))$, where we have $f_1(i') = \sum_{j \in V} \mathbb{1}(E_t(j) \leq i')p(j)$ and $f_2(v) = \max(2v-1, 0)$, effectively dropping the first half of the permuted CDF. For DIPMARK, given parameter $\alpha \in [0, 0.5]$, this is generalized by using $f_2(v) = \max(v - \alpha, 0) + \max(v - (1 - \alpha), 0)$, recovering $\gamma$-REWEIGHT for $\alpha = 0.5$. The former two variants perform detection using a log-likelihood ratio test (requiring access to $LM$), while DIPMARK uses a model-independent test.

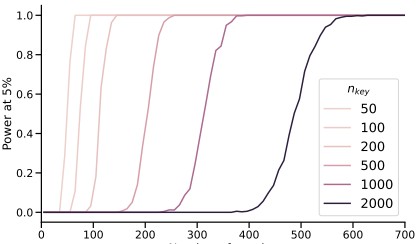 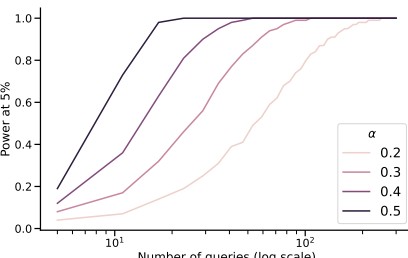

Figure 5: *Left*: power at 5% of the Fixed-Sampling test under the infinite diversity assumption. *Right*: power at 5% of the Cache-Augmented test assuming $f_1 = 0.5$. In both figures, the power is evaluated using 1000 repetitions of the test.

## E   SCALING OF THE TESTS POWER WITH THE NUMBER OF QUERIES

In this section, we study the evolution of the test's power with respect to the number of queries for both the Fixed-Sampling and the Cache-Augmented tests (§3 and §4). For the Red-Green test, we have already presented a study of the influence of the number of queries in §5.2.

**Fixed-Sampling test**   We first consider the Fixed-Sampling test (§3) under the assumption that, under the null, the model diversity is infinite and, under the alternative, that the model outputs are sampled from a uniform categorical distribution with $n_{key}$ choices. We then simulate the test for different values of $n_{key}$ and the number of queries $n$ to estimate the power. In Fig. 5 (left), we see that the test power has a quick phase transition from 0 to 1 at a number of queries less than the key size. This means that the test is robust to the increase in key size, as the watermark strength decreases linearly with $n_{key}$ according to Kuditipudi et al. (2024).

**Cache-Augmented test**   We consider the Cache-Augmented test (§4) under the assumption that the cache is cleared between queries and that $f_1 = 0.5$. We then simulate the test under those assumptions for different values of $\alpha$ and the number of queries $Q := Q_1 = Q_2$. In Fig. 5 (right), we see that for the range of $\alpha$ values considered in Wu et al. (2024), the test power reaches 1 in at most $Q = 100$ queries, which highlights the cost-effectiveness of the test.

## F   ADDITIONAL EXPERIMENTS

In this section we present several additional experiments. In App. F.1 we extend our main results to additional models and scheme variants. In App. F.2–F.6 we demonstrate the robustness of our tests to the multi-key variant of Red-Green schemes, the no-cache variant of Cache-Augmented schemes, the SynthID-Text (Google DeepMind, 2024) scheme, the adversarial modification which selectively disables the watermark, and an entropy-conditioned variant of the AAR watermark (Aaronson, 2023) that is inspired by Christ et al. (2024), respectively.

### F.1   ADDITIONAL MODELS AND SCHEME VARIATIONS

We extend the experiments from §5.1 using four additional models, LLAMA2-7B, LLAMA3-8B, YI1.5-9B and QWEN2-7B, as well as more variations of the watermarking schemes' parameters to further assess the robustness of the tests. The experimental setup for the additional results is consistent with the one described in §5.1.

Our exhaustive results are presented in Table 4. The same conclusion applies to the two additional models: the null hypothesis (*the specific watermark is not present*) is rejected at a 95% confidence level only when the corresponding watermarking scheme is applied to the model. These results confirm that the modeling assumptions for each test are satisfied across a wide range of models, indicating the tests' relevance in practical scenarios.

Moreover, in Tables 5 and 6 we present the rejection rates of our tests at 5% and 1% significance levels, respectively. These results show that the p-values are, for independent runs of the experiments, consistently low across all models and watermarking schemes when the models are watermarked and

Table 4: Additional results of our watermark detection tests across different models and watermarking schemes. We report median p-values across 100 repetitions of the experiment, and for RED-GREEN schemes additionally over 5 watermarking keys. p-values below 0.05 (test passing) are highlighted in bold. $\delta$R and $\gamma$R denote $\delta$-REWEIGHT and $\gamma$-REWEIGHT schemes, respectively.

| | | Unwatermarked | | Red-Green | | | | | | | | Fixed-Sampling | | Cache-Augmented | | |
| | | | | LeftHash | | | | SelfHash | | | | ITS/EXP | | DiPmark/$\gamma$R | | $\delta$R |
| Model | Test | $T=1.0$ | $T=0.7$ | $\delta,\gamma=2,0.25$ | $\delta,\gamma=2,0.5$ | $\delta,\gamma=4,0.25$ | $\delta,\gamma=4,0.5$ | $\delta,\gamma=2,0.25$ | $\delta,\gamma=2,0.5$ | $\delta,\gamma=4,0.25$ | $\delta,\gamma=4,0.5$ | $n=256$ | $n=2048$ | $\alpha=0.3$ | $\alpha=0.5$ | |
|---|---|---|---|---|---|---|---|---|---|---|---|---|---|---|---|---|
| MISTRAL 7B | R-G (§2) | 1.000 | 1.000 | **0.000** | **0.000** | **0.000** | **0.000** | **0.000** | **0.000** | **0.000** | **0.000** | 1.000 | 1.000 | 1.000 | 1.000 | 1.000 |
| | Fixed (§3) | 0.938 | 0.938 | 0.938 | 0.938 | 0.938 | 0.938 | 0.938 | 0.938 | 0.938 | 0.938 | **3.7e-105** | **1.8e-06** | 0.938 | 0.938 | 0.938 |
| | Cache (§4) | 0.570 | 0.667 | 0.607 | 0.639 | 0.632 | 0.608 | 1.000 | 1.000 | 0.742 | 0.824 | 0.638 | 0.687 | **2.4e-4** | **2.1e-3** | **5.6e-27** |
| LLAMA2 13B | R-G (§2) | 0.149 | 0.663 | **0.000** | **0.014** | **0.000** | **0.000** | **0.000** | **0.000** | **0.000** | **0.000** | 0.121 | 0.128 | 0.149 | 0.149 | 0.149 |
| | Fixed (§3) | 0.972 | 0.869 | 0.938 | 0.938 | 0.938 | 0.938 | 0.938 | 0.938 | 0.938 | 0.966 | **8.1e-122** | **1.5e-07** | 0.938 | 0.938 | 0.938 |
| | Cache (§4) | 0.708 | 0.573 | 0.511 | 0.596 | 0.623 | 0.807 | 0.657 | 0.619 | 0.710 | 0.583 | 0.518 | 0.692 | **1.8e-2** | **5.3e-3** | **6.7e-32** |
| LLAMA2 70B | R-G (§2) | 1.000 | 1.000 | **0.000** | **0.000** | **0.000** | **0.020** | **0.000** | **0.020** | **0.000** | **0.000** | 1.000 | 1.000 | 1.000 | 1.000 | 1.000 |
| | Fixed (§3) | 0.938 | 0.525 | 0.968 | 0.963 | 0.889 | 0.968 | 0.963 | 0.987 | 0.975 | 0.990 | **4.5e-125** | **1.7e-08** | 0.938 | 0.968 | 0.938 |
| | Cache (§4) | 0.596 | 0.620 | 0.657 | 0.797 | 0.824 | 0.639 | 0.535 | 0.651 | 0.608 | 0.593 | 0.463 | 0.818 | **1.5e-3** | **4.4e-3** | **5.8e-28** |
| LLAMA2 7B | R-G (§2) | 1.000 | 1.000 | **0.000** | **0.000** | **0.000** | **0.000** | **0.000** | **0.000** | **0.000** | **0.000** | 1.000 | 1.000 | 1.000 | 1.000 | 1.000 |
| | Fixed (§3) | 0.994 | 0.986 | 0.938 | 0.938 | 0.938 | 0.938 | 0.938 | 0.938 | 0.938 | 0.938 | **1.87e-60** | **0.003** | 0.938 | 0.938 | 0.938 |
| | Cache (§4) | 0.604 | 0.602 | 0.623 | 0.705 | 0.728 | 0.593 | 0.620 | 0.718 | 0.610 | 0.593 | 0.476 | 0.588 | **4.2e-6** | **4.5e-7** | **1.3e-21** |
| LLAMA3 8B | R-G (§2) | 1.000 | 1.000 | **0.000** | **0.000** | **0.000** | **0.000** | **0.000** | **0.000** | **0.000** | **0.000** | 1.000 | 1.000 | 1.000 | 1.000 | 1.000 |
| | Fixed (§3) | 0.938 | 0.938 | 0.938 | 0.938 | 0.938 | 0.938 | 0.938 | 0.938 | 0.968 | 0.938 | **2.3e-60** | **0.004** | 0.938 | 0.938 | 0.938 |
| | Cache (§4) | 0.734 | 0.504 | 0.605 | 0.514 | 0.712 | 0.605 | 0.600 | 0.731 | 0.729 | 0.714 | 0.618 | 0.605 | **5.2e-5** | **3.2e-8** | **3.5e-18** |
| YI1.5 9B | R-G (§2) | 0.633 | 0.194 | **0.000** | **0.000** | **0.000** | **0.000** | **0.000** | **0.000** | **0.000** | **0.000** | 0.639 | 0.620 | 0.633 | 0.633 | 0.633 |
| | Fixed (§3) | 0.938 | 0.938 | 0.938 | 0.938 | 0.968 | 0.938 | 0.952 | 0.986 | 0.986 | 0.986 | **5.8e-62** | **0.002** | 0.938 | 0.967 | 0.938 |
| | Cache (§4) | 0.609 | 0.513 | 0.609 | 0.644 | 0.716 | 0.680 | 0.619 | 0.513 | 0.705 | 0.490 | 0.618 | 0.564 | **0.000** | **0.000** | **0.000** |
| QWEN2 7B | R-G (§2) | 0.865 | 0.571 | 0.117 | **0.001** | **0.000** | **0.000** | **0.001** | **0.016** | **0.001** | 0.179 | 0.865 | 0.865 | 0.865 | 0.865 | 0.865 |
| | Fixed (§3) | 0.938 | 0.938 | 0.938 | 0.938 | 0.938 | 0.938 | 0.938 | 0.938 | 0.938 | 0.938 | **1.1e-60** | **0.002** | 0.938 | 0.938 | 0.938 |
| | Cache (§4) | 0.573 | 0.678 | 0.550 | 0.678 | 0.618 | 0.559 | 0.687 | 0.566 | 0.691 | 0.510 | 0.687 | 0.683 | **0.000** | **0.000** | **0.000** |

Table 5: Additional results of our watermark detection tests across different models and watermarking schemes. We report the rejection rate at a significance level of 5%.

| | | Unwatermarked | | Red-Green | | | | | | | | Fixed-Sampling | | Cache-Augmented | | |
| | | | | LeftHash | | | | SelfHash | | | | ITS/EXP | | DiPmark/$\gamma$R | | $\delta$R |
| Model | Test | $T=1.0$ | $T=0.7$ | $\delta,\gamma=2,0.25$ | $\delta,\gamma=2,0.5$ | $\delta,\gamma=4,0.25$ | $\delta,\gamma=4,0.5$ | $\delta,\gamma=2,0.25$ | $\delta,\gamma=2,0.5$ | $\delta,\gamma=4,0.25$ | $\delta,\gamma=4,0.5$ | $n=256$ | $n=2048$ | $\alpha=0.3$ | $\alpha=0.5$ | |
|---|---|---|---|---|---|---|---|---|---|---|---|---|---|---|---|---|
| MISTRAL 7B | R-G (§2) | 0.00 | 0.00 | **1.00** | **1.00** | **1.00** | **1.00** | **1.00** | **1.00** | **1.00** | **1.00** | 0.00 | 0.00 | 0.00 | 0.00 | 0.00 |
| | Fixed (§3) | 0.00 | 0.00 | 0.00 | 0.00 | 0.00 | 0.00 | 0.00 | 0.00 | 0.00 | 0.00 | **1.00** | **1.00** | 0.00 | 0.00 | 0.00 |
| | Cache (§4) | 0.02 | 0.04 | 0.10 | 0.00 | 0.00 | 0.04 | 0.00 | 0.00 | 0.10 | 0.04 | 0.00 | 0.06 | **0.88** | **0.80** | **1.00** |
| LLAMA2 13B | R-G (§2) | 0.05 | 0.01 | **0.98** | **0.54** | **1.00** | **1.00** | **1.00** | **1.00** | **1.00** | **0.95** | 0.05 | 0.02 | 0.05 | 0.05 | 0.05 |
| | Fixed (§3) | 0.00 | 0.00 | 0.00 | 0.00 | 0.00 | 0.00 | 0.00 | 0.00 | 0.00 | 0.00 | **1.00** | **1.00** | 0.00 | 0.00 | 0.00 |
| | Cache (§4) | 0.02 | 0.02 | 0.06 | 0.04 | 0.00 | 0.10 | 0.02 | 0.06 | 0.04 | 0.02 | 0.04 | 0.00 | **0.56** | **0.62** | **1.00** |
| LLAMA2 70B | R-G (§2) | 0.00 | 0.00 | **1.00** | **1.00** | **0.97** | **0.64** | **1.00** | **0.94** | **1.00** | **1.00** | 0.00 | 0.00 | 0.00 | 0.00 | 0.00 |
| | Fixed (§3) | 0.00 | 0.00 | 0.00 | 0.00 | 0.00 | 0.00 | 0.00 | 0.00 | 0.00 | 0.00 | **1.00** | **1.00** | 0.00 | 0.00 | 0.00 |
| | Cache (§4) | 0.02 | 0.10 | 0.02 | 0.00 | 0.04 | 0.00 | 0.04 | 0.04 | 0.06 | 0.14 | 0.02 | 0.08 | **0.78** | **0.86** | **1.00** |
| LLAMA2 7B | R-G (§2) | 0.00 | 0.00 | **1.00** | **1.00** | **1.00** | **1.00** | **1.00** | **1.00** | **1.00** | **1.00** | 0.00 | 0.00 | 0.00 | 0.00 | 0.00 |
| | Fixed (§3) | 0.00 | 0.00 | 0.00 | 0.00 | 0.00 | 0.00 | 0.00 | 0.00 | 0.00 | 0.00 | **1.00** | **1.00** | 0.00 | 0.00 | 0.00 |
| | Cache (§4) | 0.04 | 0.04 | 0.06 | 0.02 | 0.02 | 0.04 | 0.02 | 0.08 | 0.00 | 0.02 | 0.02 | 0.02 | **1.00** | **0.96** | **1.00** |
| LLAMA3 8B | R-G (§2) | 0.00 | 0.00 | **1.00** | **1.00** | **0.98** | **0.82** | **1.00** | **1.00** | **1.00** | **1.00** | 0.00 | 0.00 | 0.00 | 0.00 | 0.00 |
| | Fixed (§3) | 0.00 | 0.00 | 0.00 | 0.00 | 0.00 | 0.00 | 0.00 | 0.00 | 0.00 | 0.00 | **1.00** | **1.00** | 0.00 | 0.00 | 0.00 |
| | Cache (§4) | 0.06 | 0.04 | 0.04 | 0.06 | 0.02 | 0.06 | 0.06 | 0.04 | 0.00 | 0.04 | 0.04 | 0.04 | **1.00** | **0.98** | **1.00** |
| YI1.5 9B | R-G (§2) | 0.00 | 0.00 | **0.00** | **1.00** | **1.00** | **1.00** | **1.00** | **0.00** | **1.00** | **0.00** | 0.00 | 0.00 | 0.00 | 0.00 | 0.00 |
| | Fixed (§3) | 0.00 | 0.00 | 0.00 | 0.00 | 0.00 | 0.10 | 0.00 | 0.00 | 0.00 | 0.10 | **1.00** | **1.00** | 0.30 | 0.00 | 0.00 |
| | Cache (§4) | 0.07 | 0.06 | 0.07 | 0.05 | 0.03 | 0.05 | 0.03 | 0.03 | 0.03 | 0.02 | 0.06 | 0.06 | **0.94** | **1.00** | **1.00** |
| QWEN2 7B | R-G (§2) | 0.00 | 0.00 | **0.00** | **1.00** | **1.00** | **1.00** | **1.00** | **1.00** | **1.00** | **0.00** | 0.00 | 0.00 | 0.00 | 0.00 | 0.00 |
| | Fixed (§3) | 0.10 | 0.00 | 0.00 | 0.00 | 0.00 | 0.10 | 0.00 | 0.00 | 0.00 | 0.00 | **1.00** | **1.00** | 0.00 | 0.10 | 0.00 |
| | Cache (§4) | 0.04 | 0.01 | 0.02 | 0.04 | 0.02 | 0.01 | 0.03 | 0.04 | 0.03 | 0.02 | 0.03 | 0.01 | **0.97** | **0.97** | **1.00** |

inversely consistently high when the models are not watermarked, further confirming the reliability of our tests. For clarity, the reported rejection rates do not account for multiple testing.

## F.2 MULTIPLE KEYS IN RED-GREEN WATERMARKS

To demonstrate that the Red-Green test is robust to variations in the watermarking scheme within the same watermark family, we consider the case of Red-Green watermarks with multiple keys, where

Table 6: Additional results of our watermark detection tests across different models and watermarking schemes. We report the rejection rate at a significance level of 1%.

| | | Unwatermarked | | Red-Green | | | | | | | | Fixed-Sampling | | Cache-Augmented | | |
| | | | | LEFTHASH | | | | SELFHASH | | | | ITS/EXP | | DIPMARK/$\gamma$R | | $\delta$R |
| Model | Test | $T=$ 1.0 | $T=$ 0.7 | $\delta,\gamma=$ 2,0.25 | $\delta,\gamma=$ 2,0.5 | $\delta,\gamma=$ 4,0.25 | $\delta,\gamma=$ 4,0.5 | $\delta,\gamma=$ 2,0.25 | $\delta,\gamma=$ 2,0.5 | $\delta,\gamma=$ 4,0.25 | $\delta,\gamma=$ 4,0.5 | $n=$ 256 | $n=$ 2048 | $\alpha=$ 0.3 | $\alpha=$ 0.5 | |
|---|---|---|---|---|---|---|---|---|---|---|---|---|---|---|---|---|
| MISTRAL 7B | R-G (§2) | 0.00 | 0.00 | 1.00 | 1.00 | 1.00 | 1.00 | 1.00 | 1.00 | 1.00 | 1.00 | 0.00 | 0.00 | 0.00 | 0.00 | 0.00 |
| | Fixed (§3) | 0.00 | 0.00 | 0.00 | 0.00 | 0.00 | 0.00 | 0.00 | 0.00 | 0.00 | 0.00 | 1.00 | 1.00 | 0.00 | 0.00 | 0.00 |
| | Cache (§4) | 0.00 | 0.00 | 0.00 | 0.00 | 0.00 | 0.02 | 0.00 | 0.00 | 0.02 | 0.00 | 0.00 | 0.00 | **0.48** | **0.50** | **0.98** |
| LLAMA2 13B | R-G (§2) | 0.00 | 0.00 | 0.77 | 0.10 | 1.00 | 1.00 | 1.00 | 1.00 | 1.00 | 0.92 | 0.00 | 0.00 | 0.00 | 0.00 | 0.00 |
| | Fixed (§3) | 0.00 | 0.00 | 0.00 | 0.00 | 0.00 | 0.00 | 0.00 | 0.00 | 0.00 | 0.00 | 1.00 | 1.00 | 0.00 | 0.00 | 0.00 |
| | Cache (§4) | 0.00 | 0.00 | 0.00 | 0.00 | 0.00 | 0.00 | 0.00 | 0.00 | 0.00 | 0.00 | 0.00 | 0.00 | **0.12** | **0.12** | **0.88** |
| LLAMA2 70B | R-G (§2) | 0.00 | 0.00 | 1.00 | 1.00 | 0.93 | 0.29 | 0.97 | 0.61 | 1.00 | 1.00 | 0.00 | 0.00 | 0.00 | 0.00 | 0.00 |
| | Fixed (§3) | 0.00 | 0.00 | 0.00 | 0.00 | 0.00 | 0.00 | 0.00 | 0.00 | 0.00 | 0.00 | 1.00 | 1.00 | 0.00 | 0.00 | 0.00 |
| | Cache (§4) | 0.00 | 0.00 | 0.00 | 0.00 | 0.00 | 0.00 | 0.00 | 0.00 | 0.00 | 0.00 | 0.00 | 0.00 | **0.44** | **0.52** | **0.90** |
| LLAMA2 7B | R-G (§2) | 0.00 | 0.00 | 1.00 | 1.00 | 1.00 | 1.00 | 1.00 | 1.00 | 1.00 | 1.00 | 0.00 | 0.00 | 0.00 | 0.00 | 0.00 |
| | Fixed (§3) | 0.00 | 0.00 | 0.00 | 0.00 | 0.00 | 0.00 | 0.00 | 0.00 | 0.00 | 0.00 | 1.00 | 1.00 | 0.00 | 0.00 | 0.00 |
| | Cache (§4) | 0.00 | 0.00 | 0.00 | 0.00 | 0.00 | 0.00 | 0.00 | 0.00 | 0.00 | 0.02 | 0.00 | 0.00 | **0.80** | **0.76** | **1.00** |
| LLAMA3 8B | R-G (§2) | 0.00 | 0.00 | 1.00 | 1.00 | 0.98 | 0.76 | 1.00 | 1.00 | 1.00 | 1.00 | 0.00 | 0.00 | 0.00 | 0.00 | 0.00 |
| | Fixed (§3) | 0.00 | 0.00 | 0.00 | 0.00 | 0.00 | 0.00 | 0.00 | 0.00 | 0.00 | 0.00 | 1.00 | 1.00 | 0.00 | 0.00 | 0.00 |
| | Cache (§4) | 0.00 | 0.00 | 0.00 | 0.00 | 0.00 | 0.00 | 0.00 | 0.00 | 0.00 | 0.00 | 0.00 | 0.00 | **0.72** | **0.90** | **1.00** |
| YI1.5 9B | R-G (§2) | 0.00 | 0.03 | 1.00 | 1.00 | 1.00 | 1.00 | 1.00 | 1.00 | 1.00 | 1.00 | 0.00 | 0.00 | 0.00 | 0.00 | 0.00 |
| | Fixed (§3) | 0.00 | 0.00 | 0.00 | 0.00 | 0.00 | 0.00 | 0.00 | 0.00 | 0.00 | 0.00 | 1.00 | 1.00 | 0.10 | 0.00 | 0.00 |
| | Cache (§4) | 0.02 | 0.01 | 0.01 | 0.02 | 0.00 | 0.01 | 0.00 | 0.03 | 0.00 | 0.01 | 0.01 | 0.00 | **0.86** | **1.00** | **1.00** |
| QWEN2 7B | R-G (§2) | 0.00 | 0.00 | 0.00 | 1.00 | 1.00 | 1.00 | 1.00 | 0.00 | 1.00 | 0.00 | 0.00 | 0.00 | 0.00 | 0.00 | 0.00 |
| | Fixed (§3) | 0.00 | 0.00 | 0.00 | 0.00 | 0.00 | 0.00 | 0.00 | 0.00 | 0.00 | 0.00 | 1.00 | 1.00 | 0.00 | 0.00 | 0.00 |
| | Cache (§4) | 0.00 | 0.00 | 0.00 | 0.02 | 0.01 | 0.00 | 0.01 | 0.00 | 0.01 | 0.00 | 0.00 | 0.00 | **0.92** | **0.89** | **1.00** |

Table 7: Red-Green test for no-cache variants of Cache-Augmented schemes explored in our main experiments. We report the p-values for each model and scheme parameter combination. p-values below 0.05 (test passing) are bolded.

| LLAMA2-7B | | | LLAMA3-8B | | | MISTRAL-7B | | |
| DIPMARK/$\gamma$R | | $\delta$R | DIPMARK/$\gamma$R | | $\delta$R | DIPMARK/$\gamma$R | | $\delta$R |
| $\alpha=0.3$ | $\alpha=0.5$ | | $\alpha=0.3$ | $\alpha=0.5$ | | $\alpha=0.3$ | $\alpha=0.5$ | |
|---|---|---|---|---|---|---|---|---|
| **0.000** | **0.000** | **0.000** | **0.000** | **0.000** | **0.000** | **0.000** | **0.000** | **0.000** |

the key $\xi$ is uniformly selected from a predefined pool of keys at each generation. Using $n$ keys turns Eq. (3) into

$$l_{t_1,t_2}(x) = x^0/T + \delta''_{t_2}(x) + \varepsilon'_{t_1,t_2}(x), \tag{16}$$

with $\delta''_{t_2}(x)$ in $\{k\delta/(nT) \,|\, \forall k \in \{-n,...,n\}\}$ is obtained by averaging the variables $\delta'_{t_2}(x)$ over the set of keys. Despite modeling changes, the core assumption of logit bias being conditioned on $t_2$ remains unchanged. Therefore, we can apply the same test as in §2 to detect the watermark. Consequently, we conducted the Red-Green test on both the LeftHash and SelfHash variants using $n=3$ keys on three models (LLAMA2-13B, LLAMA2-70B and MISTRAL-7B). Recent work (Pang et al., 2024) shows that using too many keys can lead to other vulnerabilities.

Across all three models and scheme parameters, the null hypothesis (*the Red-Green watermark is not present*) is rejected at a 95% confidence level, with median p-values lower than $1e\text{-}4$ for each combination of model and setting. It shows that the Red-Green test is robust even in settings that slightly deviate from the original modeling considered in §2. It emphasizes the test's reliance on the foundational principles behind Red-Green schemes rather than on specific implementation details.

### F.3 NO-CACHE VARIANTS OF CACHE-AUGMENTED SCHEMES

The $\delta$-REWEIGHT, $\gamma$-REWEIGHT and DIPMARK schemes introduced in §4 can also be used without a cache, as suggested in (Hu et al., 2024). Interestingly, in this case these schemes belong to the Red-Green family, and the Red-Green test is applicable to detect their presence. In Table 7, we show that the watermark is indeed reliably detected across three different models. This further highlights

Table 8: Additional results for LLAMA3-8B under the adversarial modification where the watermark is disabled every $k$ queries. We report median p-values across 100 repetitions of the experiment. p-values below 0.05 (test passing) are highlighted in bold.

| | | Unwatermarked | | Red-Green | | | | Fixed-Sampling | | Cache-Augmented | | |
| | | | | LEFTHASH | | SELFHASH | | ITS/EXP | | DIPMARK/γR | | δR |
| Disable every | Test | $T = 1.0$ | $T = 0.7$ | $\delta,\gamma = 2, 0.25$ | $\delta,\gamma = 4, 0.5$ | $\delta,\gamma = 2, 0.5$ | $\delta,\gamma = 4, 0.25$ | $n_{key} = 256$ | $n_{key} = 2048$ | $\alpha = 0.3$ | $\alpha = 0.5$ | |
|---|---|---|---|---|---|---|---|---|---|---|---|---|
| 2 queries | R-G (§2) | 1.000 | 0.999 | **0.000** | **0.000** | **0.000** | **0.000** | 1.000 | 1.000 | 1.000 | 1.000 | 1.000 |
| | Fixed (§3) | 0.938 | 0.938 | 0.938 | 0.938 | 0.938 | 0.968 | **1.2e-55** | **0.001** | 0.938 | 0.938 | 0.938 |
| | Cache (§4) | 0.608 | 0.755 | 0.611 | 0.657 | 0.598 | 0.662 | 0.745 | 0.765 | **0.028** | 0.058 | **5.8e-09** |
| 3 queries | R-G (§2) | 1.000 | 1.000 | **0.000** | **0.000** | **0.000** | **0.000** | 1.000 | 1.000 | 1.000 | 1.000 | 1.000 |
| | Fixed (§3) | 0.938 | 0.938 | 0.938 | 0.938 | 0.938 | 0.938 | **2.4e-74** | **1.1e-04** | 0.938 | 0.938 | 0.938 |
| | Cache (§4) | 0.577 | 0.414 | 0.600 | 0.600 | 0.732 | 0.639 | 0.629 | 0.650 | **3.0e-04** | **1.4e-04** | **5.4e-10** |
| 5 queries | R-G (§2) | 1.000 | 0.998 | **0.000** | **0.000** | **0.000** | **0.000** | 1.000 | 1.000 | 1.000 | 1.000 | 1.000 |
| | Fixed (§3) | 0.938 | 0.938 | 0.938 | 0.938 | 0.938 | 0.938 | **8.0e-93** | **4.0e-06** | 0.938 | 0.938 | 0.938 |
| | Cache (§4) | 0.657 | 0.617 | 0.780 | 0.639 | 0.693 | 0.661 | 0.639 | 0.662 | **0.001** | **0.007** | **7.5e-17** |

Table 9: Additional results for LLAMA3-8B with the Fixed-Sampling test under the adversarial modification where the watermark is randomly disabled on every token with a fixed probability. p-values below 0.05 (test passing) are highlighted in bold.

| Model | Key size | 1% disabled | 2% disabled | 5% disabled |
|---|---|---|---|---|
| LLAMA3 8B | 256 | **9.5e-59** | **1.3e-29** | **7.3e-4** |
| | 2048 | **4.3e-4** | **3.2e-2** | 0.65 |

that the test targets a universal behavior of the watermarking scheme family and not a specific scheme instantiation.

### F.4 SYNTHID-TEXT

After the first version of our work was completed, Google DeepMind has deployed the novel SynthID-Text LLM watermarking scheme on their Gemini Web and App endpoints (Google DeepMind, 2024), and subsequently open sourced the scheme (Dathathri et al., 2024). As this represents the first real-world deployment of an LLM watermarking scheme, we use this as a case study to verify the robustness of our tests.

Namely, we notice that despite being significantly different from existing schemes and introducing several novel components such as tournament sampling, which effectively introduces a variable logit bias, the SynthID-Text scheme can still be seen as belonging to the Red-Green family from the perspective of our tests. This further demonstrates that fundamental ideas behind the schemes are commonly shared, and suggests that tests based on these ideas can often be applied to novel schemes as well. To verify this, in a new experiment, we apply the Red-Green test to the SynthID-Text scheme on the GEMMA-7B model, successfully obtaining a median p-value of $0.000$ across 100 runs. Interestingly, the only difference compared to our previous applications of the test from §2 is that the query-level hashing proposed by SynthID-Text requires us to strictly set $H = h$ instead of $H \geq h$. Therefore, we suggest using our reliable context size estimation tests (App. B.1) as a preprocessing step to enable the application of a Red-Green detection test in this case.

We have attempted to extend this experiment by repeating it on the new GEMINI 1.5 FLASH API endpoints, but we were unable to find evidence of SynthID-Text, which is in agreement with the claims made by DeepMind (Google DeepMind, 2024). Applying the test to the supposedly watermarked Web version of the model is challenging due to the need to manually produce the test data which is a tedious process—we will investigate ways to apply the test in this case further.

### F.5 ADVERSARIAL MODIFICATION: DISABLING THE WATERMARK

In this section, we provide additional results on LLAMA3-8B under two adversarial modifications to the watermarking scheme. The first adversarial modification consists of disabling the watermarking scheme every $k \in \mathbb{N}$ queries. This modification can be applied to all three watermarking families. The second adversarial modification is specific to Fixed-Sampling schemes and consists of randomly

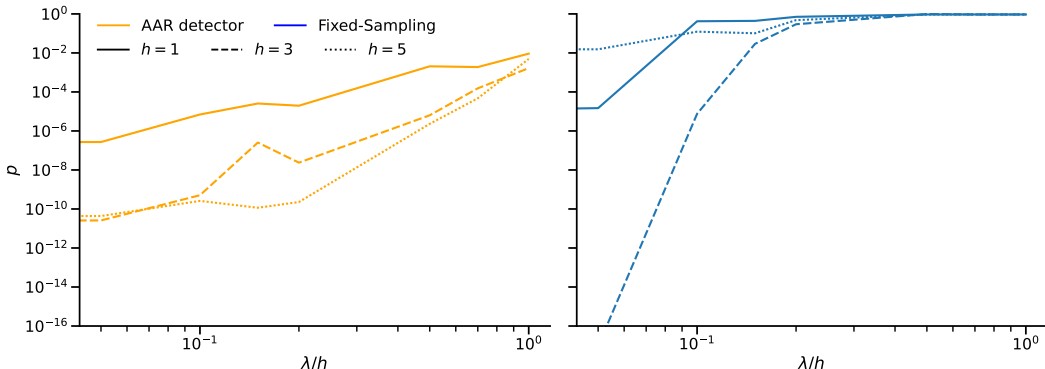

Figure 6: **Left**: Median p-values of the original watermark detector (computed with the private key $\xi$). **Right**: p-values of the Fixed-Sampling test with 50 tokens generated per query and 1000 queries.

disabling the watermark on a given token with a fixed probability for all tokens within a query. This breaks the determinism of the watermarking scheme, leading to more diverse outputs under the watermark.

We show that the tests remain relatively effective in these settings. This highlights that the tests are designed to target the fundamental mechanisms of the watermarking scheme families, and not the specific implementation details. While this shows that hiding the watermark with such modifications is not effective, we acknowledge that more sophisticated adversarial settings could reduce the effectiveness of the tests or break them entirely.

**Disabling the watermark on the query level**   In Table 8, we show the results of all tests under the adversarial modification where the watermark is disabled every 2, 3, and 5 queries. We see that despite the "dilution" of the watermark, the tests still work and have similar results as in §5. This highlights that the tests are not specific to one watermark implementation but based on fundamental mechanisms underlying each test family.

**Disabling the watermark on the token level**   In Table 9, we show the results of the Fixed-Sampling test under the adversarial modification of disabling the watermarking scheme randomly at every token. We see that the adversarial change indeed reduces the test power, but for a reasonable percentage of tokens disabled, the tests still remain effective. Moreover, as detailed in App. E, increasing the number of queries can mitigate the effect of this change.

### F.6   PRACTICAL UNDETECTABILITY: ENTROPY-CONDITIONED AAR

As introduced in §7, prior work proposed a watermark with theoretical guarantees of undetectability (Christ et al., 2024). While promising, this scheme still lacks experimental validation and current evaluations suggest it may be difficult to make it practical, mainly due to slow generation speeds (Fairoze et al., 2023). This complicates the task of evaluating our tests on this exact scheme.

**Towards practical undetectability**   Inspired by the insight in Christ et al. (2024) that conditioning the watermark on entropy improves stealth, we propose a new proof-of-concept hybrid scheme variant designed to explore the potential trade-off between the stealth and strength of watermarks.

Recall that Christ et al. (2024) generates tokens without a watermark until the generated sequence exceeds an entropy threshold; then, this high-entropy context is used to seed the scheme. Conditioning on the context's entropy ensures that the watermark is undetectable. Indeed, both our Red-Green and Fixed-Sampling tests rely on low-entropy generation to successfully detect the presence of a watermark. Therefore, to improve the tested scheme's undetectability, Red-Green schemes can be disabled if the $h$ previous tokens are below an entropy threshold. For Fixed-Sampling schemes, a similar principle can be applied where the key is used only if the previous tokens exhibit sufficiently high entropy.

---

**Algorithm 6** Entropy-Conditioned AAR Watermarking

---

**Require:** Prompt $P$, model $\mathcal{M}$, private key $\xi$, hyperparameter $\lambda$, and hash function $Hash$

1:  $x \leftarrow []$           ▷ Empty completion
2:  **for** each token $i$ to generate **do**
3:       $\{p_i\}_{i \in \Sigma} \leftarrow \mathcal{M}(P + x)$
4:       $E \leftarrow \sum_{t=i-h}^{i-1} - \log(p_t)$
5:       **if** $E > \lambda$ and $i > h$ **then**
6:           **for** each token $i \in \Sigma$ **do**
7:               $r_i \leftarrow Hash((x_{i-h}, ..., x_{i-1}), \xi)$
8:               $s_i \leftarrow r_i^{1/p_i}$
9:           **end for**
10:           $x_i^* \leftarrow \arg\max_{i \in \Sigma} s_i$
11:       **else**
12:           $x_i^* \leftarrow \text{Sample}(\{p_i\}_{i \in \Sigma})$
13:       **end if**
14:       $x \leftarrow x + [x_i^*]$
15: **end for**
16: **return** $x$

---

**Entropy-conditioned AAR**  In this preliminary investigation, we modify the *AAR watermark* proposed in Aaronson (2023). In the original scheme, the $h$ previous tokens are hashed using a private key $\xi$ to obtain a score $r_i$ uniformly distributed in $[0, 1]$ for each token $i$ in the vocabulary $\Sigma$. Given $p_i$, the original model probability for token $i$, the next token is then deterministically chosen as the token $i^*$ that maximizes $r_i^{1/p_i}$. Hence, the AAR watermark can be seen as belonging to both the Red-Green family and the Fixed-Sampling family. To make the scheme less detectable, we introduce a hyperparameter $\lambda$. Given $p_{i-h}, ..., p_{i-1}$, the probabilities of the $h$ previous tokens, the watermark is applied if and only if $\sum_{t=i-h}^{i-1} - \log(p_t) > \lambda$. We detail this new scheme in Algorithm 6.

In Fig. 6, we show, for different contexts $h$ and values of $\lambda$ (adjusted by $h$), the results of both the original watermark detector (computed using $\xi$) on the left and the results of our Fixed-Sampling test (same settings as for Table 1) on the right. To generate the samples for the original watermark detector, following the method in (Kirchenbauer et al., 2023), we generate 100 completions of 200 tokens, using prompts sampled from C4. We observe that increasing $\lambda$ reduces the strength of the watermark (left side) but also decreases the strength of our detection test at an even faster rate (right side). This suggests that using watermarking conditioned on entropy is a valid approach to make the watermark less detectable, albeit at the cost of the watermark's strength. As noted in §7, finding a satisfactory tradeoff, i.e., designing a practical scheme that is hard to detect while maintaining strength and other key properties, remains an open question. A suitable evaluation would have to evaluate FPR at low TPR, dependence on text length, as well as robustness to scrubbing. Although we believe that there is an inherent trade-off between these properties, we leave this more thorough evaluation for future work and hope that our tests can serve as a baseline to estimate empirical detectability.

## G  DETAILED COST ANALYSIS OF THE TESTS

Table 10: The costs of our watermark detection tests with the test settings used in Table 1. We assume 75 tokens per prompt in both Red-Green and Cache-Augmented tests, and a 15 tokens output. $Q, Q_1, Q_2, N$ are parameters of the tests introduced in §2–§4. The costs are estimated using current GPT4o pricing (November 2024).

| Test | Number of queries | Instantiation | Input tokens | Output tokens | Cost |
|---|---|---|---|---|---|
| Red-Green | $Q_1 \times Q_2 \times N$ | $Q_1 = 9, Q_2 = 9, N = 100$ | 648000 | 162000 | 2.43$ |
| Fixed-Sampling | $Q$ | $Q = 1000$ | 10000 | 50000 | 0.28$ |
| Cache-Augmented | $Q_1 + Q_2$ | $Q_1 = 75, Q_2 = 75$ | 10000 | 15000 | 0.10$ |

We derive the cost of each of our tests introduced in §2–§4 in Table 10.

