# OpenReview forum: "Black-Box Detection of Language Model Watermarks"
_ICLR.cc/2025/Conference — ICLR 2025 Poster_

### Official Review · Reviewer_xkZg · 2024-10-20

**Soundness:** 4
**Presentation:** 4
**Contribution:** 3
**Rating:** 8
**Confidence:** 4

**Summary:**

This paper presents a significant contribution to the field of LLM watermarking. From the authors' claims, they are the first to provide a comprehensive study of black-box watermark detection. Their findings demonstrate the practical detectability of prominent watermarking schemes, challenging previous assumptions about their undetectability. This paper has provided the foundation for future research on more robust watermarking techniques and advanced detection methods.

**Strengths:**

This paper is extremely well-written. Kudos to the authors for taking time to ensure that the paper is concise, clear, and enjoyable enough for anyone to read. The formulations for each statistical test for detectability are clear and well explained. Providing detailed tests for each class of watermarks further strengthened the paper. The results highlight the strength of their approach as watermarks can be detected accurately, more so at a low cost. I also appreciate the fact that they experimented to see if their tests could cross detect other watermarks.

**Weaknesses:**

- The methods, while detailed, appear to focus on a strict reverse engineering approach for detecting each specific class of watermark. Did the authors explore the possibility of a unified approach that could detect all classes of watermarks? What are the authors' thoughts on this?

- The experiments were limited to just three classes of watermarks. I believe this is okay, and future work could expand the scope to include other types, but it is a weakness for this paper.

- The cross-detection tests only applied to watermarks from different classes. However, there were no evaluations on whether the detection is robust to variations in the hyperparameters of the same watermark. Can the detection identify a watermark regardless of the hyperparameters used?

- Additionally, the paper lacks details on the efficiency of the detection tests. For instance, how many tokens are required to reliably detect the presence of watermarks using these methods? Addressing this could further minimize costs.

**Questions:**

My questions are outlined in the weaknesses mentioned earlier. Please address those and the following:

- In transitioning from an attack-focused approach to a defensive one, do the authors believe that their tests would still be effective in detecting the presence of watermarks in texts that have been adversarially manipulated to remove them, especially in a blackbox scenario?

**Details Of Ethics Concerns:**

The authors already states that the pros of their study outweigh the cons, and I am inclined to side with them.

---

> ### Author Response · Authors · 2024-11-21
> **Response to Reviewer xkZg**
>
> We thank the reviewer for their detailed feedback and are genuinely glad to hear that they greatly appreciate the writing of the paper. Below, we address the concerns raised in questions Q1 to Q5; and note that a revised version of the paper has been uploaded, with updates highlighted in blue. We are happy to clarify further if there are additional questions.
>
> **Q1: How much does the focus on three watermark families limit the scope of the proposed tests?**\
> Good question—our results suggest that these three families encompass a broad range of watermarks, including some released after the first version of our work (see below).
>
> First, we note that each test follows the same high-level idea: a specific behavior of a broad watermarking family is exploited to prompt the LLM with two different distributions. While this naturally raises the question of the extension to new paradigms, we make the point that targeting these fundamental properties enables us to detect a large set of currently popular and practical schemes, which either directly fall into one of these families or combine several of these properties.
>
> To substantiate this argument further, in our new revision, we provide three new experiments:
> - In Appendix A, we analyze the first public large-scale deployment of an LLM watermark, Google Deepmind’s SynthID-Text, which was open-sourced after our submission. We discuss the differences compared to previous schemes and show that our Red-Green test works perfectly on this scheme, further demonstrating its applicability.
> - In Appendix A, we consider a variant of DiPMark/$\gamma$R and $\delta$R schemes without the cache, as suggested by Rev. PSUp. Interestingly, these variants now fall into the Red-Green family, and as our new results demonstrate, are detectable by our Red-Green test.
> - In Appendix F, we consider a variant of the Aaronson watermark [5] (Figure 6), which belongs to both the Red-Green and the Fixed-Sampling families, and show that the Fixed-Sampling test can detect it for small values of its key parameter $\lambda$, including $\lambda=0$ which corresponds to the original Aaronson scheme.
>
> As we discuss in Sec. 7, while completely novel approaches can in principle always exist, our results strongly suggest that our tests are broadly applicable to most currently practical schemes.
>
> **Q2: Did the authors explore the possibility of a unified test for all watermarks?**\
> While this could be interesting, we see value specifically in having the tests be separate. In particular, we argue that specificity has the benefit of allowing for more power and more precise results, as we can directly know to which family the tested scheme belongs. The latter can help enable downstream attacks, e.g., the attacks in [1,2] are only applicable to Red-Green schemes. Similarly, knowing the family allows for parameter estimation, which is a necessary step to mount such attacks.
>
> We believe it may be possible to unify the different tests within a single prompt. However, given that the total cost of running our tests is roughly \\$3, we don’t see the practical benefits of a single unified test for the three tested families. Moreover, a joint test could be more complex, harder to understand and may not be necessarily cheaper. Finally, in case of fundamentally new scheme families, even a joint test that we hypothetically devise now would still need to be updated/revised, as it would not be directly applicable.
>
> We welcome follow-up work that improves the fundamental aspects of detection (power, detection of newer schemes), and believe that our tests can serve as a solid baseline and further provide insight into the key drawbacks of each of the fundamental ideas used in the literature from the perspective of detectability.
>
> **Q3: Are the proposed tests robust to variations in the scheme hyperparameters?**\
> We have experimentally validated our tests across a wide range of scheme  hyperparameters. More specifically, we had shown in Table 3 the results of the tests with the following hyperparameters:
> - Unwatermarked models with different temperatures.
> - Red-Green watermarks with different hash functions, $\delta$ and $\gamma$.
> - Fixed-Sampling watermarks with different length of key.
> - Cache-Augmented watermarks with different underlying schemes.
> We also tested all these parameter combinations across 7 different models.
>
> As the results are consistent across the tested hyperparameters, we believe that these variations in hyperparameters are sufficient to experimentally demonstrate the robustness of our tests. We are happy to explore more variations of parameters if the reviewer has some concrete examples.

---

> > ### Author Response · Authors · 2024-11-21
> >
> > **Q4: Could you discuss the efficiency of the test with respect to the number of tokens?**\
> > We agree with the reviewer that discussing the cost of the tests is relevant and should be clearly presented in our experimental evaluation.
> >
> > We had discussed in Appendix D how the power of the Cache-Augmented test and the Fixed-Sampling test scales with the number of queries. Additionally, we had discussed in Figure 2 how many samples should be used for the Red-Green test and how many tokens per query should be used for the Fixed-Sampling test. We then choose our test hyperparameters based on those experiments and show that running all three tests cost around $3.
> >
> > Based on the reviewer feedback, we have added a new table (Table 10) in Appendix G, which more clearly summarizes the number of tokens per test using the hyperparameters of the experiments in Table 1.
> >
> > **Q5: Do the authors believe that their tests would still be effective in detecting the presence of watermarks in texts that have been adversarially manipulated to remove the watermark, especially in a blackbox scenario?**\
> > We are not entirely certain whether we understand the reviewer's question.
> >
> > If the reviewer is referring to third-party modifications to remove the watermark (for instance, paraphrasing), since we are directly querying the model provider’s LLM, this does not affect our method. Indeed, it does not make sense for the watermark provider themselves to try to remove their own watermark. If the reviewer is referring to attempts by the model provider to hide the watermark to increase undetectability, we study such adversarial modifications in Appendix E. We are happy to follow up if our answer does not fit the reviewer's question.
> >
> > [1] “Large Language Model Watermark Stealing With Mixed Integer Programming”, Zhang et al., 2024\
> > [2] “Watermark stealing in large language models”, Jovanovic et al., ICML 2024

---

> > > ### Comment · Reviewer_xkZg · 2024-11-23
> > >
> > > Thank you for taking the time to address my queries, run additional experiments, and update the paper.
> > >
> > > Regarding Q5, my question was based on a hypothetical scenario drawn from your ethics statement: Suppose an attacker does not have access to the provider's LLM or detector but is aware that a watermark is being used. If the attacker paraphrases the text (and let's assume the paraphrased text can bypass the provider's detector), would your detector still be able to identify the watermark? My reasoning for this question is that, based on the proposed method, it intuitively seems that your approach might be more robust to such corruptions compared to the provider's detectors. However, I could be wrong, which is why I wanted to understand your perspective on this scenario (hence not putting it as a weakness).
> > >
> > > As for your other responses, I am thoroughly convinced of the potential of your method. Excellent work, and I will be increasing my scores accordingly. Kudos!

---

> > > > ### Author Response · Authors · 2024-11-23
> > > >
> > > > We thank the reviewer for their quick turnaround time and for raising their score.
> > > >
> > > > We understand Q5 now—it is quite a different scenario from the one we focus on, but nonetheless interesting. It is hard to make a conclusive statement here, but our intuition is that the way we choose prompts is crucial to our success, and instead having access to a set of ~arbitrary responses of the model might make things much more difficult. There could be ways to adapt our method to be more suitable for this case though.

---

### Official Review · Reviewer_PSUp · 2024-10-30

**Soundness:** 3
**Presentation:** 2
**Contribution:** 3
**Rating:** 6
**Confidence:** 5

**Summary:**

The authors introduce statistical tests for detecting three main watermark families under blackbox setting, namely, Red-Green, Fixed-Sampling, and Cache-Augmented watermarks. They confirm the effectiveness of their methods in an extensive experimental evaluation across seven schemes and five open-source models, and execute them on three deployed models.

**Strengths:**

This paper suggests that current watermarking schemes may be susceptible to detection in the black-box setting and verify it in their experiments.

**Weaknesses:**

- This paper lacks a clear mathematical presentation of its algorithms, and the descriptions are often vague.

- The detection tasks for Fixed-Sampling and Cache-Augmented watermarks are trivial, and the proposed simple algorithm can be easily defended against.
  1. The detection algorithm based on unique outputs is not practical. In real-world applications, one can simply skip the first few tokens to ensure that generated outputs are different, which has been proposed in Algorithm 3 in Christ et al, 2024[1].
  2. The detection algorithm focused on cache is not applicable. It could take too much time for the detection to complete in waiting for the cache to be reset in a global cache. While user cache is usually not applicable due to a potentially large number of users.
  3. The cache mechanism is only a minor component of these watermarking schemes, and removing it often does not degrade performance, as discussed in Hu et al., 2023[2].


- Reporting median p-values over 5 watermarking keys is impractical, as only a single watermarking key is typically used per model in real-world applications.

The median p-value is not a good metric, as it does not reflect the actual false positive rate. It is also difficult to interpret.

- As shown in Figure 3, there are large deviations from the actual $\delta$, indicating that the current results may not be suitable for downstream tasks.

[1] Christ, Miranda, Sam Gunn, and Or Zamir. "Undetectable watermarks for language models." The Thirty Seventh Annual Conference on Learning Theory. PMLR, 2024.

[2] Hu, Zhengmian, et al. "Unbiased watermark for large language models." arXiv preprint arXiv:2310.10669 (2023).

**Questions:**

1. The key algorithm for calculating the p-value in lines[197-240] is too vague. Could you please clarify it?

2. Could you provide a false positive rate for your detection algorithms? Additionally, the false positive rate may increase as we need to test various different types of watermarking schemes.

3. Could you provide a detailed algorithm for estimating the context size as described in lines[781-791], along with the corresponding experimental results?

---

> ### Author Response · Authors · 2024-11-21
> **Response to Reviewer PSUp**
>
> We thank the reviewer for their detailed and exhaustive feedback. Below, we address the concerns raised in questions Q1 to Q9; and note that a revised version of the paper has been uploaded, with updates highlighted in blue. We are happy to clarify further if there are additional questions.
>
> **Q1: Could the authors provide more mathematical descriptions of the algorithms for the detection test?**\
> We believe that the more descriptive introduction of each test (Sections 2, 3, and 4) better conveys the intuition behind each step and makes the material more approachable. Nonetheless, following the reviewer’s suggestions, we have added (Appendix G) algorithmic presentations of the Red-Green test, the Fixed-Sampling test and the Cache-Augmented test. The algorithm for the Red-Green test in Appendix G also addresses the lack of clarity regarding the permutation test used for computing the p-values, which was flagged by the reviewer. We are happy to improve the writing further if the reviewer has additional suggestions.
>
> **Q2: Is bypassing the Fixed-Sampling test trivial? More broadly, is there an approach to make schemes undetectable by any test?** \
> We respectfully disagree with the statement that bypassing the Fixed-Sampling test is trivial. First, as we show in Appendix E, our test works even under various adversarial modifications. Second, as we state in Section 7 (Limitations), it is possible that future schemes break our tests, and we do not believe this reduces the value of our contributions. Lastly, adversarial modifications need to be included in the broader picture of LLM watermarking: do such modifications have adversarial effects on strength, robustness to removal, or behavior in low-entropy settings?
>
> However, we agree with the reviewer that the insight from [1], conditioning the watermark on the entropy of the context, may be possible to leverage to make a given scheme less detectable. Running the adversarial modification suggested by the reviewer, and setting $\lambda$ as an entropy bound on the first previous tokens, the Fixed-Sampling test can detect this modified scheme up to $\lambda = 2$ on the Llama model. Yet, by simply rewriting the prompt to first let the model generate independent high entropy text, even with $\lambda=10$ we obtain a p-value of $2.9 \times 10^{-40}$. The test with the updated prompt does not reject the null hypothesis when the watermark is not present ($p=0.94$). This shows that the intuition behind our Fixed-Sampling test is both relevant and practical. The updated prompt is: *Write a short essay about war; but first prepend 10 random pinyin characters. Here is the format of your response: {chinese characters} {essay}. Start the essay with ##Start. Don't repeat the instructions.*.
>
> To explore the question of such modifications even further, in a newly added Appendix F, we present a new experiment analyzing a stronger extension of the scheme proposed by the reviewer. We show that modifying the scheme by conditioning the watermark on the entropy of the context bypasses our tests but also reduces the watermark strength.
>
> Namely, we used the Aaronson watermarking scheme [2] which is part of the Fixed-Sampling family, and applied the watermark on tokens where the $h$ previous tokens entropy is greater than $\lambda$ (the scheme is detailed in Algorithm 2). This means not only that the first few tokens are generated without a watermark (as in the reviewer suggested adversarial modification), but also any tokens that do not satisfy the entropy criteria. Compared to the additional reviewer suggestion, the second point prevents clever prompt engineering from bypassing the entropy mechanism to detect the watermark.
>
> We find that increasing $\lambda$ decreases the original watermark strength/robustness, but also decreases our test ability to detect the watermark at an even faster rate (Figure 6). For reference, our test succeeds up to $\lambda = 0.1$. This suggests a trade-off between watermark undetectability and watermark strength/robustness. It intuitively makes sense as, in the limit, using the scheme from [1] guarantees undetectability. We note, however, that [1] itself suffers from severe practical limitations [3] and lacks robustness evaluation. Hence, any of our tests being bypassable by some schemes is not surprising, and we don’t claim to be able to detect any possible watermarking scheme. But we remark that such modifications come at the cost of the watermark strength/robustness. Our work allows model providers or legal authorities to have realistic expectations regarding the effectiveness and pitfalls of a given watermarking scheme; and enables them to choose a scheme appropriate for their need.

---

> > ### Author Response · Authors · 2024-11-21
> >
> > **Q3: Are the explored cache implementations impractical?**\
> > We first highlight that in [4] the authors do not provide a concrete instantiation of the cache mechanism. Hence, we believe we are first to discuss concretely how a model provider could deploy the cache in practice. We consider two options that we found natural: per-user and global cache.
> >
> > We strongly disagree with the claim that a per-user cache is not practical. While we agree that model providers have a large number of users (OpenAI claims to have, on average, 200 million users per week), the cost of storing one hash table per user appears negligible compared to the cost of actually running the model (or storing the discussions history, as in the case of ChatGPT). Hence, we do not see any obvious reason why a per-user cache should not be applicable.
> >
> > Regarding the global cache, we argue that for popular models, waiting for the cache to be cleared is not a practical issue. For instance, assuming 200 million users per week and 1,000 tokens generated per user per day, this suggests that roughly 30 billion tokens are generated per day. Because with a cache there is a trade-off between cache size and watermark strength/robustness, we believe a practical instantiation of a cache would be comparatively small (as also hinted at in [4]). Hence, we argue that the cache would have to be cleared frequently enough to allow for feasible detection.
> >
> > We note that as long as no practical cache instantiation is deployed and disclosed by model providers, it is hard to make any certain statements about the real-world deployment of caches. However, given the above we do believe that our current assumptions are not inherently impractical and actually provide greater detail than some prior work in this area.
> >
> > We agree that discussing the effects of practical instantiations of the cache is important and can guide model providers. We now included in the updated paper that we are the first work to open the discussion regarding how to instantiate the cache mechanism. If the reviewer has other ideas about cache instantiations, we are happy to consider those.
> >
> > **Q4: Is the cache only a minor component of the schemes? Can the schemes from [4] and [5] be detected without the cache?**\
> > We agree with the reviewer that the cache can be seen as a modular component added on top of an already existing scheme, as we hinted on in Section 4.
> >
> > As per the reviewer's suggestion, we added in Appendix A (Table 6) the results of the Red-Green test on both $\delta$-reweight and DiPmark ([4] and [5]) without cache. Both schemes can be detected with the Red-Green test.
> >
> > More generally, we argue that (to our knowledge) all prior works proposing watermarking schemes with cache could also be instantiated without cache. Indeed, the cache is added to guarantee $b$-shot undetectability (as defined in [4]) and is not in itself a watermark. Yet, because of the cache mechanism, either the Red-Green test or the Fixed-Sampling test could fail to detect a watermark (despite the watermarking scheme behind the cache belonging to one of these families). Hence, the Cache-Augmented test ensures that those schemes can, in fact, be detected.
> >
> > **Q5: Could the authors provide FPR for their results instead of median p-values?**\
> > We chose median p-values for our main results (Table 1), as it is a common metric provided in the field ([6] and [7]) and provides more robust insights into the distribution of p-values.
> >
> > Yet, we do agree with the reviewer that providing FPR at different rejection rates is also important to gain a better understanding of the test's effectiveness. This is why we had already provided, in Table 3 and Table 4, the rejection rates at 1% and 5% for our main experiments (in particular, covering all experiments from Table 1). Those two tables show both the TPR (in the columns where the watermarking scheme agrees with the test) and the FPR (in the other columns).
> >
> > **Q6: Could you provide more detail regarding the estimation of the context size and provide additional results?**\
> > Certainly. We added in Appendix C1 a detailed description of the context size estimation algorithm. Moreover, we show in Figure 4 the distribution of the different logits estimation for different prompts and with an increasing context size.
> >
> > We are happy to improve the clarity of the estimator and provide additional results if the reviewer has additional suggestions.

---

> ### Author Response · Authors · 2024-11-21
>
> **Q7: Why is the Red-Green test reported over 5 watermarking keys when, in practice, only one key is used?**\
> We believe this is a misunderstanding. For the results presented in Table 1, the key is fixed while conducting the test. The key is simply changed between different independent repetitions of the test. The goal is to ensure that the test works no matter which private key is used by the watermark. We updated the paper to clarify this point.
>
> Further, in Appendix A, we test the case of a multiple-key Red-Green watermark (as similarly proposed in [10]), which corresponds to the case where, for each token, the key used for watermarking is randomly chosen from a fixed set of keys. However, this is an orthogonal experiment and unrelated to the 5 watermarking keys from Table 1.
>
> **Q8: Is $\delta$-estimation suitable for downstream tasks despite large errors?**\
> The main focus of our work is the detection of the watermark and demonstrating that estimating parameters at a relatively low cost is possible; however, as the reviewer notes, the estimator for $\delta$ may not be very accurate.
>
> Regarding downstream tasks, to our knowledge, there is no work that requires the knowledge of $\delta$ yet, so we cannot determine if the presented estimator is accurate enough. However, some prior works ([8,9]) required knowledge of both which watermark is present and its context size. In that case, because we experimentally achieve 100% accuracy in context size estimation (due to its discrete nature), we strongly believe that it is suitable for such downstream tasks.
>
> **Q9: Would your p-values drastically increase if you run several detection tests in sequence?** \
> As each test is performed independently (new samples are generated each time), this is an instance of the multiple testing problem.
>
> In our paper, the reported rates and p-values are presented without any multiple testing correction. Multiple testing is a field of research in its own right, and there are multiple strategies to aggregate p-values. The challenge of multiple adjustments is in defining the family of hypotheses. One scenario could be that a malicious user wants to detect only Red-Green watermarks. Another could be that a malicious user performs our three tests along with other tests of their own. In both cases, the family of hypotheses is different, and so is the way to adjust for multiple testing. Hence we report our p-values and rejection rates without accounting for multiple testing.
>
> To avoid any confusions, we updated the paper to clarify that the reported rejection rates do not account for multiple testing.
>
>
> [1] “Undetectable watermarks for language models”, Christ et al., COLT 2024\
> [2] “Watermarking of large language models”, Scott Aaronson, 2023 Workshop on Large Language Models and Transformers, Simons Institute, UC Berkeley\
> [3] “Publicly detectable watermarking for language models”, Fairoze et al., 2024\
> [4] “Unbiased watermark for large language models”, Hu et al., ICLR 2024\
> [5] “Dipmark: A stealthy, efficient and resilient watermark for large language models”, Wu et al., ICML 2024\
> [6] “Robust distortion-free watermarks for language models”, Kuditipudi et al., TMLR 05/2024\
> [7] “On the learnability of watermarks for language models”, Gu et al., ICLR 2024\
> [8] “Large Language Model Watermark Stealing With Mixed Integer Programming”, Zhang et al., 2024\
> [9] “Watermark stealing in large language models”, Jovanovic et al., ICML 2024\
> [10] “A watermark for large language models”, Kirchenbauer et al., ICML 2023

---

> > ### Author Response · Authors · 2024-11-25
> > **Discussion window ending**
> >
> > We kindly remind the reviewer to let us know if our response addressed their concerns, as the discussion window ends shortly. We are happy to discuss any outstanding points further.

---

> > > ### Comment · Reviewer_PSUp · 2024-11-25
> > >
> > > I would like to thank the authors for their detailed rebuttal. Most of my concerns have been addressed, and the paper has greatly improved after the revision. I will raise my score.

---

### Official Review · Reviewer_Ay59 · 2024-11-03

**Soundness:** 3
**Presentation:** 3
**Contribution:** 3
**Rating:** 6
**Confidence:** 4

**Summary:**

This paper proposes a black-box detection method for identifying whether a watermark is embedded in a Large Language Model (LLM). In this paper, the detectability of current watermarking schemes is investigated for the first time in a practical black-box environment. The researchers developed statistical test methods to detect the presence of watermarks and estimate parameters using a limited number of black-box queries for three popular families of watermarking schemes; Red-Green, Fixed-Sampling and Cache-Augmented. Experimental results show that these approaches are effective and cost-efficient across multiple open source models and different settings. The paper also discusses the ethical implications of its work, highlighting the benefits of raising awareness of the ease of detection of watermarking schemes, despite the potential risk of misuse.

**Strengths:**

1. This paper, for the first time, examines the detectability of current watermarking schemes in a practical black-box setting, which is practical in the real detection scenario.
2. The method is well written and the method makes sense and is easily understood. Each method has a clear section structure.
3. The experimental results in the black-box scenario verify the effectiveness of the method.

**Weaknesses:**

1. Although the authors pointed out that their motivation is to study the ability of current watermarks to resist detection, they did not highlight the significance of watermark detection in real scenarios. Providing specific application scenarios of black-box watermark detection can help readers better understand the contribution of black-box watermark detection.

2. The results in Table 1 indicate the method in the paper is constrained by the need for distinct detection techniques for various watermarking methods, with poor generalization among them. As more watermarking methods are proposed, this may increase the cost of detecting watermarks.

3. Minor concern: Watermark detection results in Table 2 for production-level language models accessed via API are suboptimal and you can not conclude on the presence of a watermark,  which brings some concerns to readers about real-world detection.

**Questions:**

1.	Can you discuss more application scenarios of watermark detection? This question has a great impact on the contribution of the paper.

2.	Can you discuss potential commonalities between their detection techniques for different watermarking families? The universality of watermark detection technology is beneficial to reducing detection costs.

3.	Can the authors discuss more about why current detection methods are unable to determine the watermarking method of real-world production-level LLMs?

---

> ### Author Response · Authors · 2024-11-21
> **Response to Reviewer Ay59**
>
> We thank the reviewer for their detailed feedback. Below, we address the concerns raised in questions Q1 to Q4; and note that a revised version of the paper has been uploaded, with updates highlighted in blue. We are happy to clarify further if there are additional questions.
>
> **Q1: Can you highlight the practical implications of watermark detection?**\
> Certainly. The objective behind watermarking an LLM is to enable the detection of whether a given text was generated by a specific LLM. In practice, it should allow both holding a model provider accountable for harmful text generated by its model and holding users accountable for using an LLM in scenarios where its use is inappropriate or forbidden. Being able to detect a watermark behind an LLM deployment provides a malicious user with multiple opportunities.
>
> First, detection is a common prerequisite for performing spoofing attacks [1, 2, 3, 4], where a malicious user learns the watermark in order to generate arbitrary watermarked text without using the watermarked model. Such attacks can be used to discredit a model provider by generating text that appears to be genuinely watermarked and attributing it to the model provider.
>
> Second, detection is a prerequisite for assisted scrubbing attacks (as in [1, 4]), where a malicious user can more successfully remove the watermark from an LLM generated text compared to blindly rewriting the watermarked texts. Consequently, such malicious users can nullify any positive effects associated with the watermark deployment.
>
> Lastly, knowing that a particular LLM is watermarked may lead a malicious user to avoid using that LLM entirely and instead favor another LLM that is not known to be watermarked.
>
> Hence, knowing how detectable schemes are in practice, besides theoretical interest, is also important for model providers or legal authorities to have realistic expectations regarding the effectiveness and pitfalls of a given watermarking scheme. We have added a discussion about the practical implications of watermark detection in the updated version of the paper in a newly added Appendix I referenced from our Introduction.
>
> **Q2: Does the need for one test per scheme family limit the applicability of the proposed tests?**\
> Good question—we do not believe this is the case.
>
> First, we note that each test follows the same high-level idea: a specific behavior of a broad watermarking family is exploited to prompt the LLM with two different distributions. If the distributions are highly dissimilar, it suggests that a watermark is present. Otherwise, the model is likely not watermarked. This idea is instantiated to three common paradigms: the key based on the context (Red-Green schemes), the key permutation (Fixed-Sampling) and the presence or absence of a cache (Cache-Augmented). While this naturally raises the question of the ease of extension to new paradigms, we make the point that targeting these fundamental properties enables us to detect a large set of currently popular and practical schemes, which either directly fall into one of these families or combine several of these properties.
>
> To substantiate this argument further, in our new revision, we provide three new experiments:
> - In Appendix A, we consider a variant of DiPMark/$\gamma$R and $\delta$R schemes without the cache, as suggested by Rev. PSUp. Interestingly, these variants now fall into the Red-Green family, and as our new results demonstrate, are detectable by our Red-Green test.
> - In Appendix A, we analyze the first public large-scale deployment of an LLM watermark, Google Deepmind’s SynthID-Text, which was open-sourced after our submission. We discuss the differences compared to previous schemes and show that our Red-Green test works perfectly on this scheme, further demonstrating its applicability.
> - In Appendix F, we consider a variant of the Aaronson watermark [5] (Figure 6), which belongs to both the Red-Green and the Fixed-Sampling families, and show that the Fixed-Sampling test can detect it for small values of its key parameter $\lambda$, including $\lambda=0$ which corresponds to the original Aaronson scheme.
>
> As we discuss in Sec. 7, while completely novel approaches can in principle always exist, our results strongly suggest that our tests are broadly applicable to most currently practical schemes.

---

> > ### Author Response · Authors · 2024-11-21
> >
> > **Q3: Could there be a single test for all watermarks?**\
> > As discussed in Q2, while they share the same idea on a meta level, our tests are specifically instantiated to three core ideas behind most current schemes. We argue that specificity has the benefit of allowing for more power and more precise results, as we can directly know to which family the tested scheme belongs. The latter can help enable downstream attacks, e.g., the attacks in [1,2] are only applicable to Red-Green schemes. Similarly, knowing the family allows for parameter estimation, which is a necessary step to mount such attacks.
> >
> > We believe it may be possible to unify the different tests within a single prompt. However, given that the total cost of running our tests is roughly \\$3, we don’t see the practical benefits of a single unified test for the three tested families. Moreover, a joint test could be more complex, harder to understand and may not be necessarily cheaper. Finally, in case of fundamentally new scheme families, even a joint test that we hypothetically devise now would still need to be updated/revised, as it would not be directly applicable.
> >
> > We welcome follow-up work that improves the fundamental aspects of detection (power, detection of newer schemes), and believe that our tests can serve as a solid baseline and further provide insight into the key drawbacks of each of the fundamental ideas used in the literature from the perspective of detectability.
> >
> > **Q4: Why do the tests fail on current black box LLMs?**\
> > While the results of our tests do not provide any guarantees in that regard, we believe this is because the tested APIs were indeed not watermarked—thus, we do not see this as a weakness of our work. In our new experiment in App. A we also repeat our Gemini test on the new Gemini 1.5 Flash API, and still find no evidence of a watermark. This matches the public claims of Google DeepMind, which have announced the watermark only on Web and App deployments. Note that in another new experiment in App. A we demonstrate that the same watermark can be detected by our tests when deployed locally.
> >
> > [1] “Large Language Model Watermark Stealing With Mixed Integer Programming”, Zhang et al., 2024\
> > [2] “Watermark stealing in large language models”, Jovanovic et al., ICML 2024\
> > [3] “On the learnability of watermarks for language models”, Gu et al., ICLR 2024\
> > [4] “De-mark: Watermark Removal in Large Language Models”, Chen et al., 2024\
> > [5] “Watermarking of large language models”, Scott Aaronson, 2023 Workshop on Large Language Models and Transformers, Simons Institute, UC Berkeley

---

> > > ### Comment · Reviewer_Ay59 · 2024-11-25
> > > **Raise my rating**
> > >
> > > As mentioned above, the black box detection techniques of various watermarking methods are discussed, but these are not systematically integrated, resulting in a looser overall structure and a reading experience that is closer to a blog or technical report than an academic paper. Nonetheless, the paper performs well in terms of experimentation and is well documented, so I would like to raise my score.

---

> > > > ### Author Response · Authors · 2024-11-25
> > > >
> > > > We thank the reviewer for raising their score to recommend acceptance, and appreciate the comments. We are discussing ideas to tune the writing to make it more cohesive, e.g., by adding a short section before our tests are introduced, which would give more context around detectability and the threat model. If the reviewer has other concrete suggestions, we are happy to take those into account.

---

### Official Review · Reviewer_gicP · 2024-11-04

**Soundness:** 3
**Presentation:** 4
**Contribution:** 3
**Rating:** 8
**Confidence:** 3

**Summary:**

The paper shows that it is possible to detect the presence of most existing watermarks using black-box interaction with the model, without knowing the watermarking key.
They also demonstrate that their attack is capable of estimating the parameters used in the watermarking schemes.

**Strengths:**

A huge number of watermarking papers have come out recently.
Many of them ask whether their watermarks harm generation quality by performing experimental evaluations, but these are inherently limited: There is no way to experimentally guarantee that the watermark will preserve the quality under *every possible* use-case of the model.
Therefore, perhaps a more useful test of quality is to simply attempt to detect it. If attacks that are specifically designed to detect the watermark still fail to do so, then this can be seen as unusually strong evidence that it is quality-preserving.

This work shows that existing schemes typically fall short in this respect, demonstrating an important weakness.

**Weaknesses:**

It is not surprising that they were able to easily detect the schemes they attacked. Those schemes are not designed to be undetectable.
In the "Limitations" section, they justify the choice to only consider these schemes with the claim that the provably-undetectable schemes "lack experimental validation" and "are not yet practical due to slow generation speed."

However, I believe these claims require justification because:
- "Excuse me, sir? Your language model is leaking (information)" is a practical implementation of an undetectable scheme. The author doesn't report any issues. This seems to already contradict the above claims.
- As I understand it, the generation speed of these techniques (including the one just mentioned) is _no slower_ than it is for any other scheme. They work essentially identically to other schemes, except that they are careful not to embed bias in cases where it might be noticeable without the key.
- I think that the reason there are relatively few practical demonstrations of undetectable schemes is just that most people doing experiments don't care about it. If you can get slightly better robustness by dropping undetectability, most experimentalists will go for that. However, since the message of the present paper depends on it _actually being difficult_ to build a practical undetectable scheme, it would be much more compelling if you at least attempt to do so.

Here is a simple undetectable scheme that you could try as a benchmark: Use Aaronson's scheme exactly (implemented in many places, e.g. Piet et al.), except that if a $k$-gram has empirical entropy (as defined in Christ et al.) less than $\lambda$, then don't use the Gumbel-max trick and instead just sample without bias according to the model. (Crucially, the first $k$ tokens in any response should be sampled exactly according to the model, without any watermark bias.) Note that this scheme is no slower than any other scheme. Detection with the key is also extremely fast.

It is easy to see that this scheme will require seeing roughly $2^{\lambda/2}$ tokens before it becomes detectable _without_ the key; and it should be detectable _with_ the key as long as the text has (empirical) entropy at least $\lambda$ in most sequences of $k$ consecutive tokens.
- If you find that this scheme only becomes practically undetectable once you set $k$ or $\lambda$ to be unreasonably large (such that detection with the key significantly suffers), then I would find the message that existing practical schemes much more compelling.
- If you find that this scheme is in fact practically undetectable for reasonable choices of $k$ and $\lambda$, then that would arguably be an even more compelling result (although the message would change slightly).

**Questions:**

In Appendix C, you discuss a method for estimating scheme parameters. Are your techniques capable of learning the watermarking key itself?

---

> ### Author Response · Authors · 2024-11-21
> **Response to Reviewer gicP**
>
> We thank the reviewer for their detailed feedback. Below, we address the concerns raised in questions Q1 to Q3; and note that a revised version of the paper has been uploaded, with updates highlighted in blue. We are happy to clarify further if there are additional questions.
>
> **Q1: Can you justify the claim that the scheme from [1] lacks practical validation? Does the same apply to [3]?**\
> Our claim regarding the inefficiency of [1] originates from Figure 3 in [2] (this citation was unfortunately missing before, and we added it to our Sec. 7). In particular, they demonstrate that using [1] reduces the generation speed of the model approximately 10x compared to the case without the watermark, which is prohibitive to any practical deployment. Their results suggest that the slow generation is caused by the conversion from tokens to a binary vocabulary. It is our understanding that [3] also employs the same conversion, and thus likely experiences the same issues. We did not find a corresponding latency evaluation of [3] that would contradict this. Further, both [1] and [3], amongst other properties also lack any evaluation of the watermark robustness, a property central to most LLM watermarking works. Notably, the authors of [3] acknowledge that their approach is primarily theoretical and that the given practical instantiation only serves as a proof of concept.
>
> More broadly, we greatly appreciate the attempts to construct theoretically-grounded schemes such as [1] and [3], and believe they bring us closer to understanding the difficulty of building a practical undetectable scheme. We look forward to future work on validation of such schemes, but highlight that in the current state it is hard to know their practical value, e.g., if the robustness properties are only slightly or fundamentally different from other schemes. Thus, our goal was to test our method on as many practically demonstrated schemes as possible. This is further extended by our new experiments in App. A (variants of Cache-Augmented schemes, SynthID-Text) and App. F (the reviewer’s proposed Aaronson variant, discussed below in Q2). We have updated the limitations section to better reflect our position, and are happy to make further changes if the reviewer has concrete suggestions.
>
> **Q2: Can you investigate the strength-detectability tradeoff of a variant of the AAR scheme that gets partially disabled based on entropy, inspired by the ideas of [1]?**\
> We thank the reviewer for this idea. While we already had included experiments to test the robustness of our test in adversarial settings (Appendix E), coming up with new adversarial schemes based on the idea from [1] indeed strengthens the discussion regarding the state of watermark detectability. In a new experiment, we implement and evaluate the proposed variant, presenting detailed results and a discussion in Appendix F.
>
> In summary, this change remains detectable by our method until $\lambda/k = 0.1$. We also show that the strength of the watermark decreases with $\lambda$. Hence, there is a trade-off between undetectability and watermark strength. While this partial evaluation seems promising, including other important properties of the watermark (e.g., FPR at low TPR, robustness to different entropy settings, dependence on text length, robustness to watermark removal) may reduce the viable range of parameters further, as is generally the case for LLM watermark evaluations. On the other hand, more targeted detection methods may be more effective against this variant.
>
> We included a summary of the consequences of this new finding in our Sec. 7 as a pointer for future work on finding practical undetectable schemes, and we are happy to adapt the message there further.
>
>
> **Q3: Can your parameter estimation techniques be used to learn the private key of the watermark?**\
> No—this is a much harder problem that was explored in prior work. While learning the exact key is effectively not possible for the schemes we consider, learning the full effect of the key has been shown to be possible in some instances. For example, for the Unigram scheme proposed in [4] (a Red-Green scheme with a fixed Red and Green vocabulary independent of the context), [5] proposes a method to almost perfectly recover the Red/Green partition.
>
> More generally, the field of watermark spoofing studies how to generate watermarked text without knowledge of the private key. Such spoofing attacks [6, 7, 8] only need to acquire partial knowledge of the effect of the watermark (for instance, partial knowledge of the Red/Green partition given a context) to be successful. Hence, most attacks on watermarking schemes do not rely on having full knowledge of the private key. However, as they often require some knowledge of the scheme parameters (e.g., context size), they could benefit from our parameter estimation as the first step in a more elaborate exploit of a scheme [6, 8]; chaining of attacks is an interesting future work item.

---

> ### Author Response · Authors · 2024-11-21
>
> [1] “Undetectable watermarks for language models”, Christ et al., COLT 2024\
> [2] “Publicly detectable watermarking for language models”, Fairoze et al., 2024\
> [3] “Excuse me, sir? Your language model is leaking (information)”, Zamir et al., 2024\
> [4] “Provable robust watermarking for ai-generated text”, Zhao et al., ICLR 2024\
> [5] “Large Language Model Watermark Stealing With Mixed Integer Programming”, Zhang et al., 2024\
> [6] “Watermark stealing in large language models”, Jovanovic et al., ICML 2024\
> [7] “On the learnability of watermarks for language models”, Gu et al., ICLR 2024\
> [8] “De-mark: Watermark Removal in Large Language Models”, Chen et al., 2024

---

### Author Response · Authors · 2024-11-21
**General Response**

We thank the reviewers for their feedback and evaluation of our work. We are pleased to see that they believe our contributions fill an important gap in LLM research (gicP, xkZg), and serve as a strong foundation for future studies in the field (xkZg). We are also glad they appreciate the extensiveness of our experimental evaluation (Ay59, PSUp, xkZg). We have uploaded an updated version of the paper (new content marked blue) and replied to all reviewers’ questions in individual comments below. We are happy to engage in follow-up discussions.

---

### Meta-Review · Area_Chair_bWoS · 2024-12-19

**Metareview:**

Summary: This paper studies the problem of detecting LLM watermarks in a black-box way, without even knowing the watermark key. Extensive experiments across three families of LLM watermarks, Red-Green, Fixed-Sampling and Cache-Augmented, verify the effectiveness of the proposed method.

Strengths:
1. This paper is the first work that detects LLM watermarks in a black-box way. It suggests that current watermarks are susceptible to detection in the black-box setting.
2. The paper is well-written and the experiments are rich.

Weaknesses:
1. Reviewers have concerns on the claim that the provably-undetectable schemes "lack experimental validation" and "are not yet practical due to slow generation speed."
2. Reviewers have concerns on the generalizability of the proposed detectors, as they are mostly based on reverse engineering. The experiments were limited to just three classes of watermarks.

All reviewers consistently vote for acceptance, two of who champion the paper with a score of 8. There is no doubt that the paper is above the acceptance bar of ICLR.

**Additional Comments On Reviewer Discussion:**

All reviewers consistently vote for acceptance, two of who champion the paper with a score of 8. There is no doubt that the paper is above the acceptance bar of ICLR.

---

### Decision · Program_Chairs · 2025-01-22

Accept (Poster)